# Continuous exchange of an inner-membrane ring component is required for assembly and function of the type III secretion system

Corentin Brianceau[1,2], Stephan Wimmi [1,6], Thales Kronenberger [3,4,5] & Andreas Diepold [1,2] ✉

Major bacterial pathogens manipulate eukaryotic target cells by injecting effector proteins through type III secretion systems (T3SS). Recent in situ observations revealed that these large molecular machines, often referred to as injectisomes, are remarkably dynamic and adaptive entities, with the cytosolic T3SS components forming a mobile network that recruits effectors to the export machinery. In contrast to these soluble components, the transmembrane rings anchoring the injectisome are stably associated – with one exception. Using functional assays, live cell microscopy, and photobleaching experiments, we found that SctD, which constitutes the inner membrane ring of the T3SS, exchanges subunits in secreting injectisomes in *Yersinia enterocolitica*. To elucidate the biological significance of this unexpected dynamic behavior of a key structural component, we investigated its role in the assembly and function of the T3SS. Using engineered SctD variants whose exchange rate can be modulated, we found that exchange supports the integration of export apparatus components into assembled membrane rings and efficient secretion of effectors. Our findings uncover a new aspect of the molecular function and regulation of the T3SS, which may apply to other secretion systems and molecular machines.

Many Gram-negative bacteria use type III secretion systems (T3SS) to manipulate the behavior of eukaryotic cells by injecting effector proteins into their target cells in a one-step mechanism[1–4]. The T3SS, commonly known as the injectisome, is essential for the virulence of important pathogens, including *Salmonella, Shigella, Yersinia*, but is also employed for symbiosis, mainly with insects and plants[5–7]. While the set of secreted effectors is adapted to the lifestyle of the respective bacteria, the overall structure and composition of the injectisome itself are highly conserved[8–11]: A core structure spanning both bacterial membranes anchors a cytosolic complex on the proximal side, an export apparatus in the inner membrane (IM) and a hollow needle on the distal side[12–14] (Fig. 1a). The membrane-spanning structural core of the injectisome consists of three components: the secretin protein SctC, which forms a stable 15-mer ring in the outer membrane (OM) and interacts with the bitopic IM protein SctD, organized in a 24-mer ring embedding the lipoprotein SctJ, which forms another 24-mer ring[15,16]. These components and the export apparatus are the first substructures to form during the assembly of the T3SS[17]. In the gastrointestinal pathogen *Yersinia enterocolitica*, T3SS components are expressed at 37 °C, marking the transition from environmental to host body temperature. Upon expression, the membrane rings assemble in the absence of any other injectisome component, nucleated by the

[1]Max Planck Institute for Terrestrial Microbiology, Department of Ecophysiology, Marburg, Germany. [2]Department of Applied Biology, Institute of Applied Biosciences, Karlsruhe Institute of Technology (KIT), Karlsruhe, Germany. [3]German Center for Infection Research (DZIF), partner-site Tübingen, Tübingen, Germany. [4]Institute of Medical Microbiology and Hygiene, Interfaculty Institute of Microbiology and Infection Medicine (IMIT), University of Tübingen, Tübingen, Germany. [5]School of Pharmacy, Faculty of Health Sciences, University of Eastern Finland, Kuopio, Finland. [6]Present address: Institute for Biological Physics, University of Cologne, Cologne, Germany. ✉e-mail: andreas.diepold@kit.edu

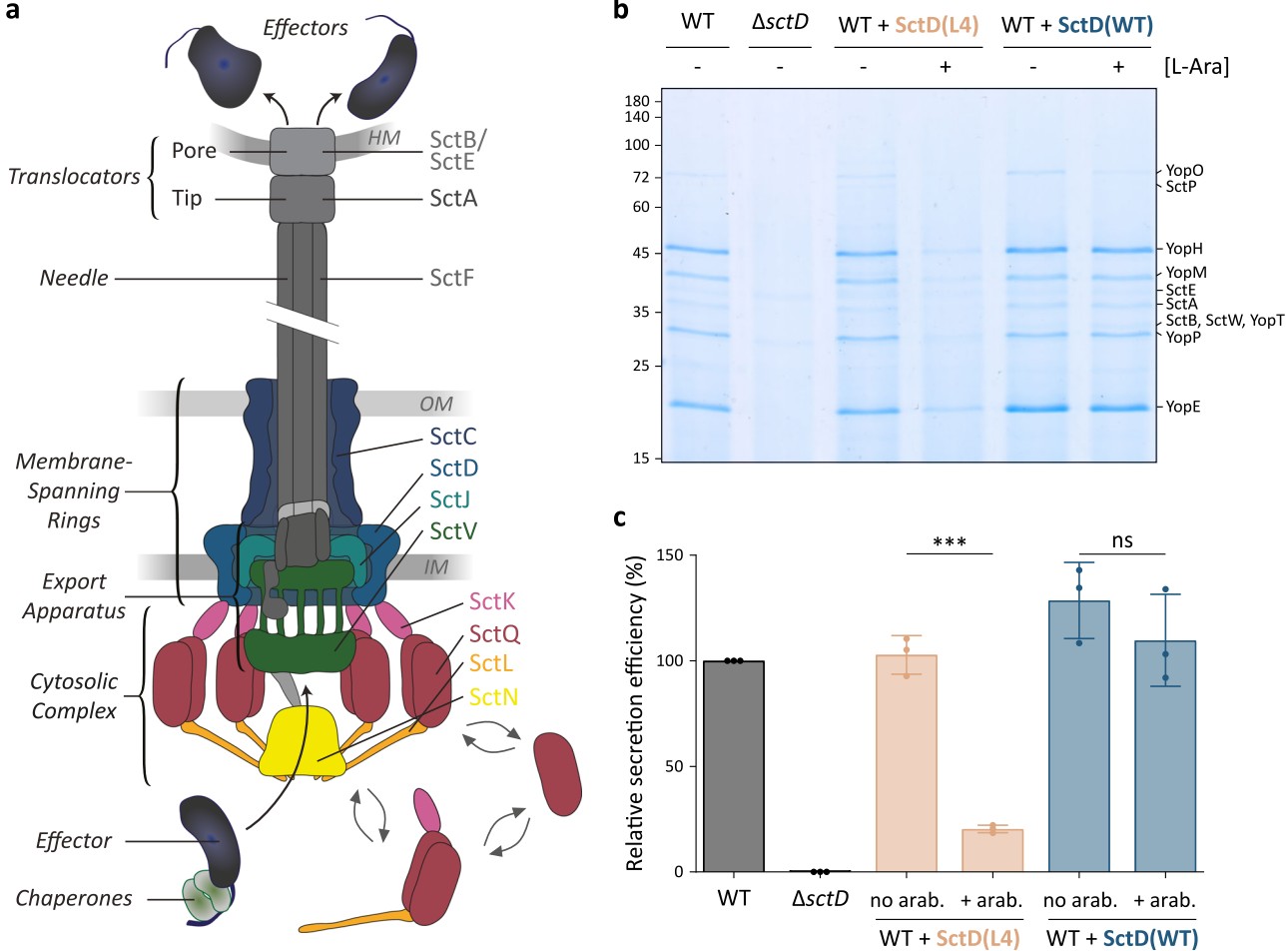

**Fig. 1 | Subunits of the inner membrane ring component SctD can be replaced in fully assembled injectisomes. a** Schematic representation of the T3SS injectisome illustrating effector export. Left, main substructures of the injectisome; right, components important for this study. HM, OM, IM: Host, bacterial outer, inner membrane. **b** Secretion assay showing the export of native T3SS export substrates (Yop = *Yersinia* T3SS effectors, Sct = structural or regulatory components) into the culture supernatant. The experiment setup and conditions for the experiment in this and all other figures of this manuscript are visualized in Supplementary Fig. 3. Expression of SctD(WT) (blue) and SctD(L4) (light salmon) was either induced by 0.01% L-arabinose (+) or repressed by the addition of 0.2% L-glucose (−).

Normalized amounts of supernatant corresponding to $3 \times 10^8$ bacteria per lane were loaded on an SDS-PAGE gel and stained with Coomassie Brilliant Blue. Representative image, $n = 3$ independent experiments. Left, molecular weight in kDa; right, names of exported proteins. **c** Quantification of relative secretion efficiency of the different strains shown in (**b**). The secretion efficiency was determined by gel densitometry for the YopO, YopH, YopM, SctA, YopP, and YopE bands and compared to the wild-type strain. Circles indicate single data points; bars show mean values, error bars depict standard deviation. ns, $p = 0.3103$; ***, $p = 0.0001$ in a two-sided homoscedastic $t$ test.

formation of the stable secretin ring spanning the OM and the peptidoglycan layer, and progressing through SctD to SctJ[18]. Similarly, the export apparatus, consisting of a pseudohelical core structure formed by $SctR_5S_4TU$ and a nonameric ring of SctV[19–22], assembles independently in the IM, where it diffuses freely within the membrane in the absence of SctCDJ[23,24]. SctJ is thought to preassemble around the export apparatus, connecting it to SctCDJ and thereby anchoring the export apparatus in the peptidoglycan[19,21,24,25] (Fig. 1a). How the large export apparatus, composed of over 70 transmembrane helices, integrates into the pre-existing membrane-spanning ring structure during T3SS assembly remains unclear.

Recent studies uncovered that the cytosolic components of the injectisome, SctK, -Q, -L, and -N, form six "pod-like" structures connected to SctD on one side and the export apparatus on the other side[26–28]. The main component of the pods, SctQ, constantly exchanges subunits between an injectisome-bound state and a cytosolic pool[29]. In the cytosol, SctQ interacts with SctK, -L, and -N to form soluble pod complexes[30], which act as effector shuttles, binding effector proteins in the cytosol and delivering them to the injectisome[31]. Accordingly, the removal of the cytosolic pool of these proteins stops secretion,

which was exploited to create a light-controlled T3SS by targeted optogenetic sequestration of the dynamic cytosolic component SctQ[32].

In contrast to SctQ, the membrane-bound components SctC (the OM secretin) and SctV (the main component of the IM export apparatus) were stably associated with the injectisome in secreting bacteria[29]. This observation suggested that the core structure of the injectisome, consisting of the membrane rings and the export apparatus, is static. This is in line with the finding that needle complexes, containing the membrane rings and the needle, can be copurified and visualized by transmission electron microscopy[15,25,33]. However, a recent study revealed that at low external pH – mimicking the environment of the stomach through which gastrointestinal pathogens must transit – the IM ring protein SctD partly dissociated from the injectisome in *Y. enterocolitica*. This partial unbinding of SctD led to a reversible dissociation of the cytosolic components and a temporary suppression of secretion under these conditions[34]. Consistent with this, single-particle tracking and live microscopy revealed that the fraction of mobile SctD molecules increased from 8% at pH 7 to 48% at pH 4, and that the major export apparatus

component SctV was released from the injectisome under these conditions[34].

Given the presence of an unbound SctD pool under pH-neutral conditions, we asked whether SctD can dynamically exchange between an injectisome-bound and a freely diffusing state in secreting bacteria. We further considered whether such dynamics might be essential for injectisome assembly – particularly for the integration of the export apparatus – or for effector secretion. In this study, we investigated these questions by employing functional assays, live cell microscopy, and fluorescence recovery after photobleaching (FRAP), revealing that SctD indeed exchanges in functional injectisomes. To investigate the physiological role of SctD exchange, we engineered an SctD variant whose exchange can be controlled by crosslinking, irreversibly by the addition of a chemical crosslinker, or reversibly depending on the redox conditions in the periplasm. Using this tool, we uncovered that SctD exchange supports the assembly and function of the injectisome, allowing the integration of the export apparatus into the SctD ring structure. Our data identifies the integral IM protein SctD, a core component of the injectisome, as a dynamic component whose exchange is vital for the T3SS. These findings provide important insights into both the assembly and function of the T3SS, highlighting the dynamics and adaptability of molecular machines.

## Results

### SctD can be replaced by a dominant-negative variant after assembly into functional injectisomes

Recent studies suggest that SctD monomers forming the IM membrane ring of the injectisome are not as static as previously thought[9,17]. At low external pH, these monomers exhibit a reversible partial dissociation from the injectisome. We therefore wondered whether SctD subunits exchange between the injectisome and an IM pool under physiological conditions, similar to the behavior of cytosolic T3SS components[34]. To investigate this question, we tested if wild-type SctD, expressed from its native locus and incorporated into pre-assembled functional injectisomes, could be replaced by a dominant-negative SctD mutant expressed from a plasmid after injectisome assembly. Previous studies have shown that the mutant SctD(L4), which contains four alanine substitutions in the cytosolic forkhead-associated (FHA) domain of SctD (Supplementary Fig. 1), exerts a dominant-negative effect on type III secretion when coexpressed with native *sctD*[35]. We decided to use this dominant-negative effect on secretion as an indicator for the incorporation of the mutant SctD(L4) after injectisome assembly. We therefore expressed either SctD(WT) or SctD(L4) from a plasmid in a wild-type strain containing the native *sctD* gene on the plasmid for *Yersinia* virulence (pYV). Cultures were first incubated at 37 °C in non-secreting conditions (presence of CaCl₂ in the medium) to enable expression of the native SctD protein and pre-assembly of functional injectisomes, while preventing the expression of the plasmid-encoded gene. Subsequently, the cultures were moved to 28 °C to repress the temperature-dependent expression of further injectisome components and the assembly of new injectisomes. Expression of SctD(WT) or SctD(L4) from the plasmid in trans was then induced by adding L-arabinose to the cultures (Supplementary Fig. 2), and effector secretion, which also occurs at 28 °C in the presence of injectisomes[36], was activated by chelation of Ca²⁺ to assess the function of the injectisome. The WT strain and the secretion-deficient Δ*sctD* mutant were used as positive and negative controls, respectively. Expression of additional SctD(L4) after injectisome assembly strongly reduced the secretion efficiency (to about 20%; Fig. 1b, c). In contrast, additional expression of SctD(WT) after injectisome assembly did not cause any significant reduction in secretion. These results suggest that SctD(WT) was replaced by the dominant negative SctD(L4) mutant in pre-assembled functional injectisomes.

To further investigate this exchange and visualize it on a single-cell basis, we examined whether non-fluorescent SctD(WT) or SctD(L4) expressed after assembly could integrate into fully pre-assembled injectisomes containing only fluorescently labeled SctD (Fig. 2a). As observed in previous studies[18,34], EGFP-SctD, expressed from the native *sctD* locus, forms stable fluorescent foci at the cell periphery, corresponding to its incorporation into assembled injectisomes (Fig. 2b). When non-fluorescent SctD(WT) or SctD(L4) were expressed from plasmid after injectisome assembly, leading to a strongly increased overall concentration of SctD (Supplementary Fig. 2), EGFP-SctD foci became less numerous and less intense, suggesting that in both conditions, the additional non-fluorescent SctD molecules replaced the initially bound EGFP-SctD molecules, leading to fewer EGFP-SctD per injectisome (Fig. 2b, d). This phenotype resulted from the additional expression of SctD, as the number and intensity of EGFP-SctD foci were similar to the positive control both prior to induction of SctD(WT) and SctD(L4) expression and when expression was repressed by glucose addition (Fig. 2b, d). These results indicate that fully assembled SctD, forming the IM ring structure of functional injectisomes, is replaced by additional freshly expressed SctD in live bacteria at native conditions.

The FHA domain in the cytoplasmic N-terminus of SctD is important for its interaction with the cytosolic T3SS component SctK[35], which in turn interacts with SctQ, -L, and -N, forming the cytosolic complex of the injectisome[12,30]. Accordingly, EGFP-SctQ, as part of the cytosolic complex, requires SctD for binding to the injectisome and localization in fluorescent foci at the bacterial membrane[18,30,34]. Because this critical FHA domain is altered in the SctD(L4) mutant, we hypothesized that its incorporation may disrupt the binding of cytosolic components to the T3SS, which would explain its dominant-negative effect on secretion. As an additional test for the exchange of SctD in pre-assembled injectisomes, we therefore tested the influence of SctD(L4) expression after injectisome assembly on the localization of EGFP-SctQ. This later expression of SctD(L4) abolished the binding of EGFP-SctQ and resulted in its relocalization into the cytosol, similar to the Δ*sctD* negative control (Fig. 2c, e). These results confirm that SctD(L4) is integrated into the IM ring of the injectisomes, preventing the binding of SctQ via SctK. This phenotype was not caused by a defect in injectisome assembly in this strain, as localization of EGFP-SctQ in foci was similar to that of the WT strain prior to SctD(L4) expression (Fig. 2c). As expected, EGFP-SctQ foci were not influenced by the expression of additional SctD(WT) (Fig. 2c, e). Taken together, these results strongly indicate that SctD is a dynamic component of the injectisome, exchanging subunits even in fully assembled injectisomes.

### Characterization and kinetics of SctD exchange

To characterize the exchange of SctD at the injectisome in more detail, we quantified the fluorescence recovery after photobleaching (FRAP) of EGFP-SctD, expressed from its native locus in the pYV plasmid. Following incubation at 37 °C to induce expression and assembly of the *Yersinia* injectisome under secreting conditions, individual EGFP-SctD foci at the cell periphery were photobleached by brief pulses of intense laser light, and their recovery, indicating the replacement of bleached with unbleached cellular EGFP-SctD proteins, was followed over time. To ensure the accuracy of SctD exchange analysis, the cytosolic component SctQ and the main component of the export apparatus SctV, known to be dynamic and static, respectively[29], were used as positive and negative controls. Remarkably, EGFP-SctD foci showed a distinct recovery after photobleaching, similar to EGFP-SctQ, whereas SctV-EGFP foci did not or very slowly reappear (Fig. 3a). The average fluorescence recovery curves showed that the exchange rate of EGFP-SctD foci was intermediate between EGFP-SctQ and SctV-EGFP foci (Fig. 3b). Quantification of individual fluorescence recovery curves yielded an average half-time recovery ($t_{1/2}$) for EGFP-SctD of

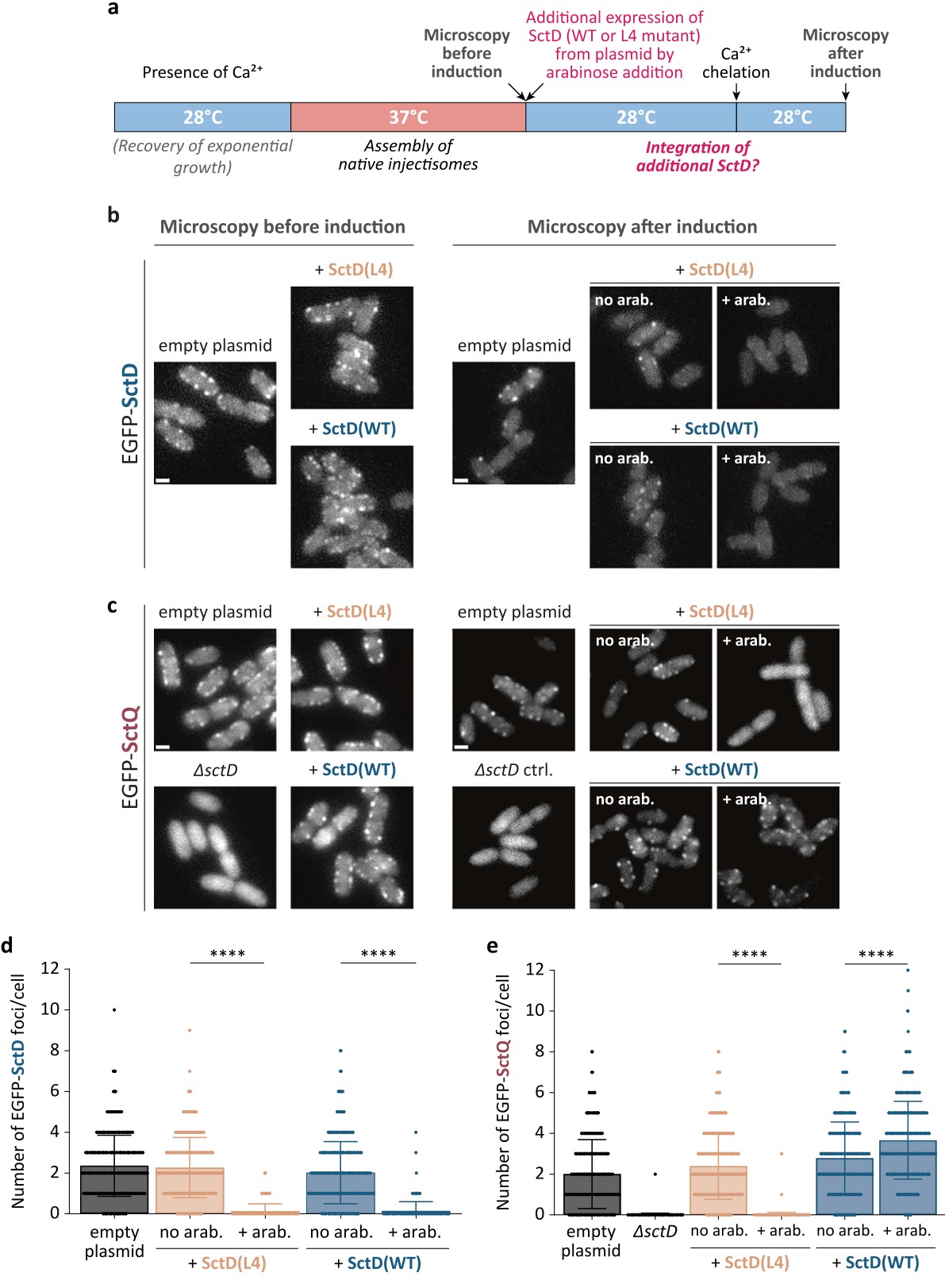

176.5 ± 13.1 s, slower than that for EGFP-SctQ ($t_{1/2}$ of 56.2 ± 4.7 s), but faster than for SctV-EGFP ($t_{1/2}$ of 401.8 ± 60.8 s) (Fig. 3c). Fluorescence recovery curves and half-time recovery values were highly reproducible across replicates (Supplementary Fig. 4). Given that the fluorescence recovery is orders of magnitude faster than the production of new T3SS components (Supplementary Fig. 5), this recovery indicates exchange of SctD monomers at the assembled injectisome, rather than de novo assembly of new injectisomes. Consequently, these results confirm the previous observation that SctD does not form a stable static ring within the injectisome, but instead exchanges subunits with kinetics slower than the known dynamic cytosolic component SctQ.

**Fig. 2 | Exchange of SctD in functional injectisomes results in dilution of EGFP-SctD fluorescence and suppression of SctQ binding upon integration of a dominant negative variant. a** Schematic timeline of the experiment displaying the timepoints at which the micrographs were taken. **b** SctD expressed from plasmid replaced EGFP-SctD in previously assembled injectisomes. Micrographs of *Y. enterocolitica* expressing EGFP-SctD from its native promoter on the virulence plasmid. Assembled injectisomes contain EGFP-SctD (Microscopy before induction, left). Subsequently, expression of additional SctD(WT) (blue) or SctD(L4) (light salmon) in trans was induced by 0.01% L-arabinose ("+ arab.") or not (addition of 0.2% glucose, "no arab."). The additional expression of both SctD variants leads to integration of these non-fluorescent SctD into the existing injectisome structures, resulting in a strong reduction of fluorescence intensity compared to the empty plasmid control. **c** Exchange of EGFP-SctD by SctD(L4), but not by SctD(WT) prevents the binding of SctQ to the injectisome. Micrographs of *Y. enterocolitica* expressing EGFP-SctQ from its native promoter on the virulence plasmid. Assembled injectisomes contain EGFP-SctQ (Microscopy before induction, left).

Subsequently, expression of additional SctD(WT) (blue) or SctD(L4) (light salmon) in trans was induced by 0.01% L-arabinose ("+ arab.") or not (addition of 0.2% glucose, "no arab."). Additional expression of the SctD(L4) variant, which cannot bind SctQ, leads to loss of EGFP-SctQ binding to the injectisomes, in line with its integration into existing injectisome structures. In contrast, additional expression and integration of SctD(WT) does not lead to loss of EGFP-SctQ binding to the injectisomes. Empty plasmid control and a Δ*sctD* strain with empty plasmid serve as controls. **b** and **c** show representative images from *n* = 3 independent experiments. Scale bars, 1 μm. **d**, **e** Quantification of the number of EGFP-SctD (d) or EGFP-SctQ (e) foci per cell in the different strain backgrounds shown in (**b**) and (**d**), respectively, after induction of expression (*n* = 3 with a total of 600 cells). Bars and error bars depict mean and standard deviation, respectively. Circles indicate single data points. ****, *p* < 0.0001 with $1.3 \times 10^{-184}$ (EGFP-SctD + SctD(L4)), $1.6 \times 10^{-142}$ (EGFP-SctD + SctD(WT)), $2.8 \times 10^{-205}$ (EGFP-SctQ + SctD(L4)) and $3.9 \times 10^{-17}$ (EGFP-SctQ + SctD(WT)) against the respective "no arab." vs "+ arab." conditions in a two-sided homoscedastic test.

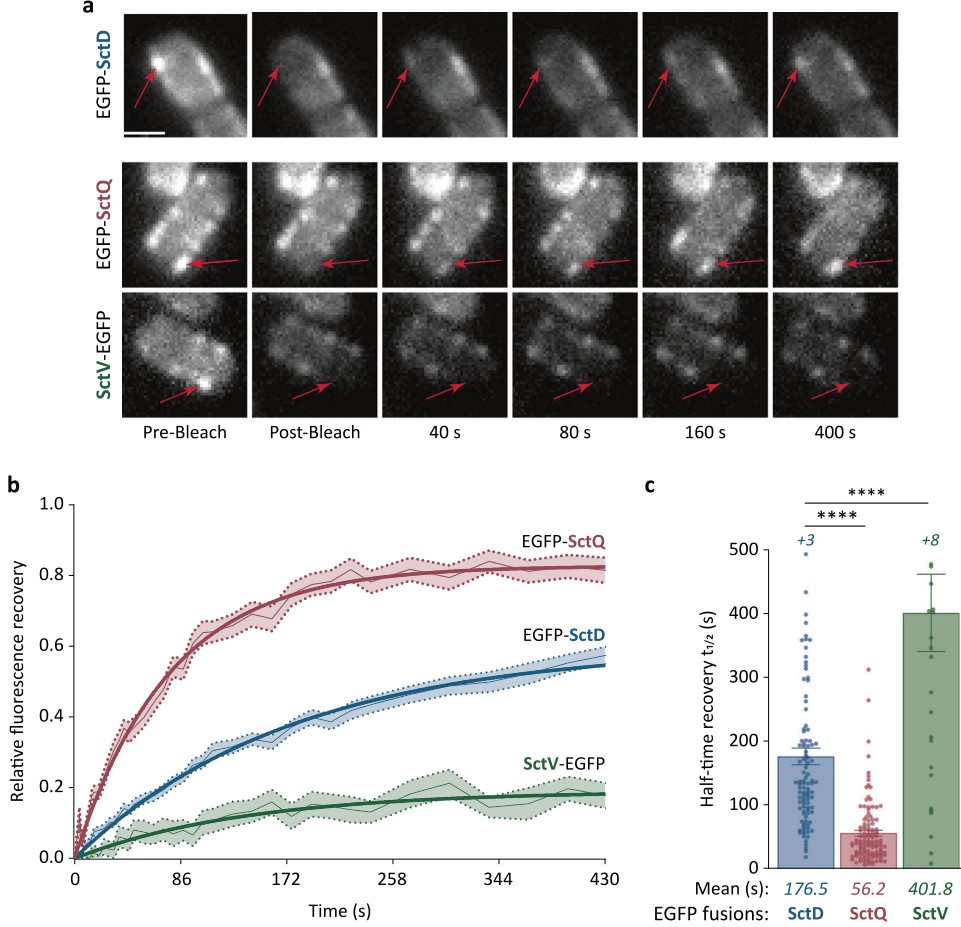

**Fig. 3 | Fluorescence recovery after photobleaching confirms and quantifies SctD exchange. a** Representative micrographs of cells expressing EGFP-SctD, SctV-EGFP, or EGFP-SctQ from the native genetic loci under secreting conditions, before and after photobleaching, as indicated. Red arrows indicate photobleached fluorescent foci. Scale bar, 1 μm. **b** Average fluorescence recovery curves (light connecting lines) with standard errors of the mean (shaded areas) and exponential regression curves (thick lines) for the indicated protein. *n* = 132 for EGFP-SctQ, 130 for EGFP-SctD, and 122 for SctV-EGFP, from 3 independent experiments. **c** Analysis of half-time recoveries of individual foci. The half-time recovery data for each strain were extracted from all individual fluorescence recovery curves with an $R^2$ value above 0.4 for exponential regression. *n* = 104 for EGFP-SctD, 113 for EGFP-SctQ, and 29 for SctV-EGFP, from 3 independent experiments. +*n*, additional values outside displayed y range (>500 s). Error bars represent the standard error of the mean. ****, *p* < 0.0001 with $2.0 \times 10^{-16}$ (EGFP-SctD vs EGFP-SctQ) and $1.4 \times 10^{-7}$ (EGFP-SctD vs SctV-EGFP) in a two-sided homoscedastic *t* test.

## Substitution of amino acids R161 and E179 in the periplasmic domain of SctD by cysteines allows for crosslinking of neighboring monomers

To directly assess the role of protein exchange in T3SS function, it is essential to be able to modulate the exchange. Earlier studies identifying dynamic components of the T3SS[29] or other exchanging proteins in bacteria[37] did not allow for such exchange modulation. However, the good structural characterization of SctD[15,38–41] inspired us to create a reversibly cross-linkable SctD variant, based on the introduction of cysteine residues at the interface between neighboring

monomers. SctD monomers contain three native cysteines, all of which are in its cytoplasmic domain, which in most prokaryotes is a reducing environment and therefore less prone to disulfide bond formation (Supplementary Fig. 1). In contrast, the periplasmic domain of SctD contains no native cysteines. We therefore hypothesized that introducing cysteines into the periplasmic domain might allow sufficiently close cysteines to form covalent disulfide bonds, which could be modulated by the addition of the reducing agent dithiothreitol (DTT)[42]. As an alternative, we selected a homobifunctional chemical crosslinker, 1,8-bismaleimido-diethyleneglycol (BM(PEG)$_2$). This reagent can cross the outer membrane but not the inner membrane[43], allowing it to irreversibly crosslink neighboring cysteine residues in the periplasm.

For the identification of suitable pairs of amino acids, we generated a model for the periplasmic portion of the *Y. enterocolitica* SctCDJ complex (*Ye*SctCDJ). A *Ye*SctCDJ model was generated based on the structure of the *Salmonella* SPI-1 injectisome (PDB ID 6PEM[15]) with a defined SctC:D:J interface. We then simulated a slice of four subunits of this complex (Fig. 4a, b) using classical molecular dynamics (based on ref. 22, see Methods for details) and analyzed the most frequent interactions in the protein-protein interfaces. Our analysis revealed a sharp conformational transition where the SctD subunits move towards each other without affecting the SctC/SctD interface. From this model, the two amino acid residues R161 and E179 were identified as being close between neighboring SctD monomers and therefore a target of choice for crosslinking (Fig. 4c, d, Supplementary Fig. 6).

To assess the possibility of modulating SctD exchange by crosslinking neighboring SctD monomers into the ring, R161 and E179 were substituted with cysteines, creating the SctD(2C) mutant (Supplementary Fig. 1). We first expressed either FLAG-SctD(WT) or FLAG-SctD(2C) from an inducible plasmid in a Δ*sctD* strain. Expression of SctD was induced at 37 °C together with the other T3SS components in secreting medium and in the presence of DTT to prevent the formation of disulfide bonds during injectisome assembly. After completion of assembly, bacteria were transferred to fresh secreting medium at 28 °C to stop both expression of SctD and other injectisome components. At this temperature, the cultures were incubated for 3 hours, either in reductive conditions (presence of DTT), in natively oxidative periplasmic conditions allowing the formation of reversible natural crosslinks, or in the presence of the chemical crosslinker BM(PEG)$_2$, irreversibly crosslinking the cysteine residues.

Western blot analysis of the cellular proteins showed a band at approximately 45 kDa, consistent with the size of SctD monomers (46.9 kDa) (Fig. 4e), for all strains and conditions. Additionally, two higher molecular weight bands were observed specifically for FLAG-SctD(2C): one at approximately 140 kDa and another heavier than 180 kDa. In the presence of DTT, the largest molecular weight band was absent, and the ~140 kDa band was significantly reduced in intensity. Conversely, treatment with BM(PEG)$_2$ increased the intensity of the higher molecular weight bands. Immunoblots against SctC and SctJ did not indicate an involvement of either of these proteins in the high-molecular-weight bands (Supplementary Fig. 7), in line with the large distance of the native cysteine residues in these two proteins (Supplementary Fig. 6f). These findings indicate that indeed, neighboring SctD(2C) monomers can be connected reversibly by the formation of disulfide bonds in the oxidative periplasmic environment or irreversibly by chemical crosslinkers. In line with this interpretation, analysis of the fractions shown in Fig. 4e using a reductive SDS-PAGE gel all but abolished the crosslinking of SctD(2C), except in the chemically crosslinked fraction (Supplementary Fig. 8).

## Reversible crosslinking allows to modulate the exchange rate of SctD(2C)

Since the previous results indicated that the crosslinking state of SctD(2C) could be modulated by the redox state of the periplasm, we

investigated whether this would influence the exchange of SctD. Cys-161 and −179 were introduced in an EGFP-labeled SctD on the pYV to analyze the exchange by FRAP. As previously observed, EGFP-SctD(WT) foci exhibited recovery of fluorescence after photobleaching (Supplementary Fig. 9). In contrast, recovery of EGFP-SctD(2C) foci was strongly reduced in native oxidative periplasmic conditions (ox.), indicating slower protein exchange due to crosslinking (Fig. 5a; Supplementary Fig. 9). In line with this observation, the half-time recovery significantly differed between the strains: EGFP-SctD(WT) foci had a $t_{1/2}$ of 202.8 ± 13.3 s, whereas EGFP-SctD(2C) foci recovered much slower with a $t_{1/2}$ of 857.6 ± 118.6 s (Fig. 5b). Addition of the chemical crosslinking agent BM(PEG)$_2$ (crossl.) did not result in a further reduction of the exchange (Supplementary Figs. 9, 10).

We next investigated whether addition of DTT, which prevents crosslinking by creating a reductive periplasmic environment (red.), restores the faster recovery of fluorescence of EGFP-SctD(2C). Indeed, while recovery of EGFP-SctD(WT) foci was only slightly accelerated upon the addition of DTT, the effect was much more pronounced for EGFP-SctD(2C). The recovery half-time ($t_{1/2}$) decreased from 857.6 ± 118.6 s to 193.6 ± 15.3 s, similar to the values observed for EGFP-SctD(WT) (Fig. 5; Supplementary Fig. 9). These findings suggest that crosslinking of SctD(2C) reversibly inhibits SctD protein exchange, depending on the redox state of the periplasmic environment. Importantly, this modulation provides a valuable tool to study the role of SctD exchange in type III secretion.

## Reduction of SctD exchange impedes T3SS protein secretion

With the confirmation that SctD(2C) exchange can be modulated, we wondered whether this reduction in exchange would impact T3SS functionality. Similar to previous experiments, bacteria expressing SctD(WT) or SctD(2C) from plasmid in a Δ*sctD* background were incubated at 37 °C in non-secreting medium. After assembly, the bacteria were shifted to 28 °C to halt injectisome assembly and then transferred to fresh secreting medium to induce the export of the effectors. Throughout the experiment, bacteria were cultivated either in reducing or native oxidative periplasmic conditions (presence or absence of DTT, respectively) to modulate the exchange of SctD(2C). In the natural oxidative environment (ox.), T3SS secretion was strongly reduced in bacteria expressing SctD(2C), which exchanges slowly under these conditions, but not in bacteria expressing SctD(WT) (Fig. 6). In contrast, under reductive periplasmic conditions (red.), where SctD(2C) exchanges normally, the secretion of the effectors by the strain expressing SctD(2C) was similar to that of the strain expressing SctD(WT). Thus, these results show that SctD exchange is directly correlated with the function of the injectisome, effector secretion, indicating a direct causative relationship of SctD exchange and injectisome function.

## SctD exchange influences assembly and function of the injectisome

Given the impact of the reduced exchange rate of SctD(2C) on protein secretion by the T3SS, we investigated whether the exchange of SctD affects T3SS assembly or the secretion process itself. To discern which aspect was affected, we quantified effector export when exchange of SctD(2C), expressed from its native genetic background, was reduced by oxidative crosslinking during the assembly and/or the secretion process (Fig. 7a). Secretion of effectors was decreased when SctD exchange was reduced by oxidative conditions during both assembly and secretion (Fig. 7b). To more precisely quantify effector export efficiency using Western blot analysis, we introduced an inducible plasmid encoding for a FLAG-tagged variant of YopE, one of the effectors secreted by the injectisomes of *Yersinia*. This approach further excludes the positive feedback effect of secretion on the expression of native promoters[44]. As observed for the native T3SS substrates, YopE-FLAG export was reduced when SctD exchange was

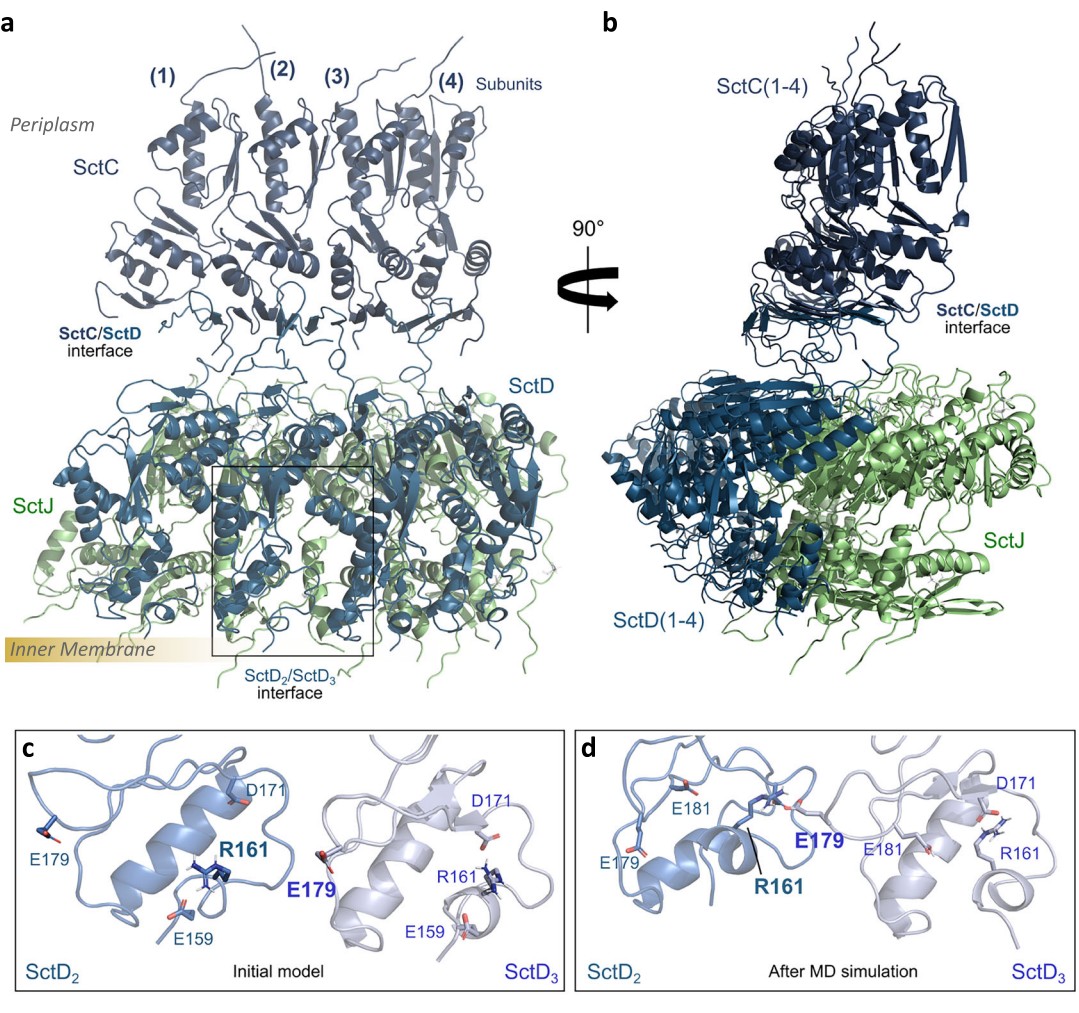

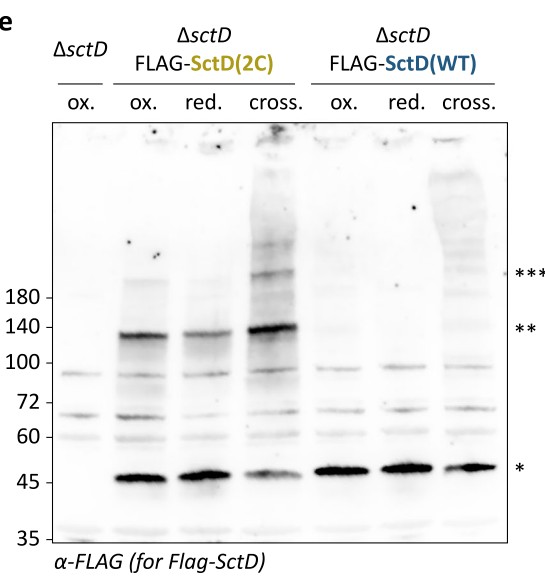

**Fig. 4 | Introduction of two proximal cysteine residues in the periplasmic domain of SctD enables the crosslinking of neighboring monomers.** Overview of the *Ye*SctCDJ model highlighting four repeated subunits of SctC$_{35-172}$ (dark blue), SctD$_{158-383}$ (turquoise) and SctJ (green) from the "front" (**a**) and "side" viewpoints (**b**). Interface between SctD subunits highlighted by a square and further depicted in (**c**, **d**). The bar on the left indicates the approximate position of the inner membrane. SctD2/SctD3 interface in the initial model (**c**) and after the molecular dynamics simulations (**d**). Simulations were run for 3×500 ns and representative frames were retrieved from hierarchical clustering using the protein backbone position variation.

**e** Western Blot anti-FLAG of total cellular protein from 1.5 ×10$^8$ bacteria expressing either FLAG-SctD(2C) or FLAG-SctD(WT) from a plasmid in a Δ*sctD* background (*n* = 3). The expression of the different SctD variants was induced using 0.03% L-arabinose. The cultures were incubated at 28 °C either untreated (ox.), in the presence of 2 mM of DTT (red.), or with 0.1 mM of BM(PEG)$_2$ (crossl.). To preserve the redox-sensitive crosslinks during the analysis, cells were resuspended in SDS sample buffer without DTT. Left, molecular weight in kDa; right, assignment of different bands observed on the Western Blot, *: SctD monomer (45 kDa); **: Band running at ~140 kDa molecular weight; ***: High molecular weight band.

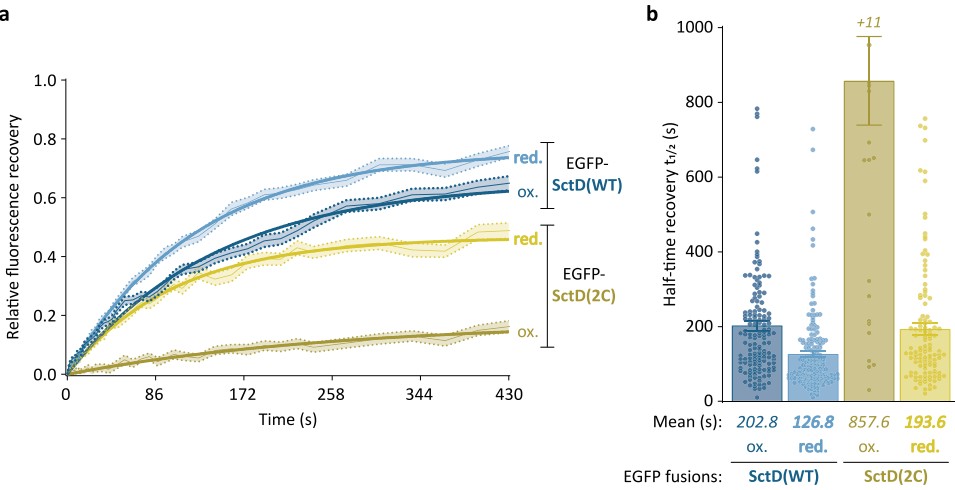

**Fig. 5 | Oxidative crosslinking of SctD(2C) modulates its exchange at the injectisome. a** Fluorescence recovery curves representing the merged data from multiple foci for the indicated strains and conditions. red./ox. = reducing/oxidizing periplasmic environment (presence/absence of 2 mM DTT). Average fluorescence recovery curves (light connecting lines) are depicted with their standard errors of the mean (shaded areas) and exponential regression curves (thick lines). $n = 181$ for EGFP-SctD(WT) red., 168 for EGFP-SctD(WT) ox., 179 for EGFP-SctD(2C) red., and 170 for EGFP-SctD(2C) ox., from 3 independent experiments. **b** Analysis of the half-time recoveries of the foci. The half-time recovery data for each strain were extracted from all individual fluorescence recovery curves with an $R^2$ value above 0.4 for exponential regression. $n = 148$ for EGFP-SctD(WT) ox., 170 for EGFP-SctD(WT) red., 29 for EGFP-SctD(2C) ox., and 116 for EGFP-SctD(2C) red. $+n$, additional values outside displayed y range (>1000 s). Error bars represent the standard error of the mean.

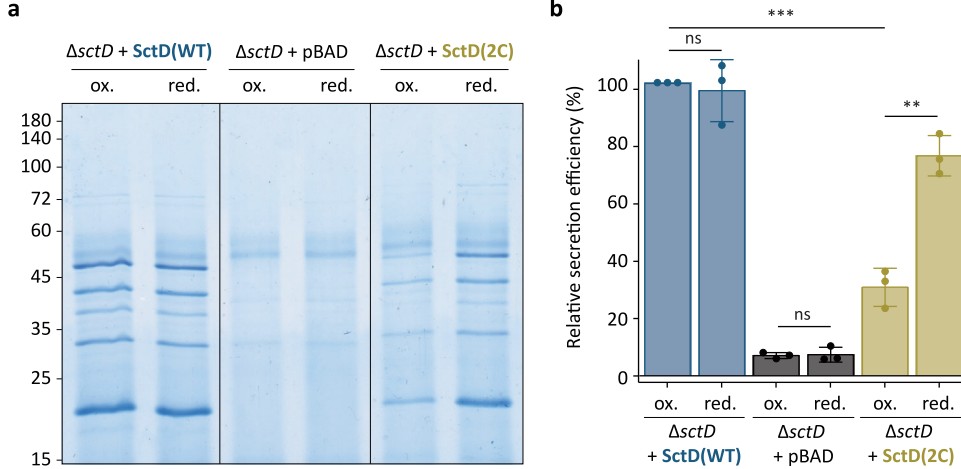

**Fig. 6 | Reducing the SctD exchange rate decreases effector secretion. a** Representative secretion assay showing the export of native injectisome substrates into the culture supernatant at 28 °C in the indicated conditions (ox., native oxidative periplasmic conditions; red., in presence of 2 mM DTT) from $n = 3$ independent experiments. Expression of SctD(WT) or SctD(2C) was induced from pBAD-based plasmids during the assembly of the injectisome by adding 0.03% of L-arabinose in non-secreting medium. After completion of the assembly, the bacteria were transferred to secreting medium to induce the secretion of the native injectisome substrates in the supernatant. Normalized amounts of supernatant corresponding to $3 \times 10^8$ bacteria per lane were loaded on an SDS-PAGE gel and stained with Coomassie Brilliant Blue. Left, molecular weight in kDa.
**b** Quantification of relative secretion efficiency of the different strains shown in (**a**). The secretion efficiency was determined by gel densitometry for the YopO, YopH, YopM, SctA, YopP, and YopE bands and compared to the wild-type complementation at oxidative conditions. Circles indicate single data points; bars show mean values, error bars depict standard deviation. ns, $p = 0.6853$ ($\Delta sctD$ + SctD(WT) "ox." vs "red.") and $p = 0.8322$ ($\Delta sctD$ + pBAD "ox." vs "red."); **, $p = 0.0012$; ***, $p = 5.1 \times 10^{-5}$ in a two-sided homoscedastic $t$ test.

prevented during T3SS assembly or secretion. The effect was stronger during secretion than during assembly, and the strongest when exchange was prevented throughout (Fig. 7b, c). These results indicate that reduced SctD exchange leads to a notable decrease in effector export during injectisome assembly and function, suggesting that SctD exchange is relevant for these processes.

## SctD exchange is not required for the exchange of the cytosolic component SctQ

The clear decrease in effector export when SctD(2C) exchange was reduced after the completion of injectisome assembly suggested a crucial role of SctD exchange in the secretion process. One conceivable pathway is a direct effect of SctD exchange on the exchange of SctQ. This dynamic cytosolic T3SS component exchanges at the injectisome faster under secreting conditions[29] and was found to recruit and shuttle effectors to the injectisome[31]. Given that SctD is essential for the binding and the localization to the injectisome of SctQ and other cytosolic components via SctK[18,30], we therefore tested whether SctD exchange influences the exchange of SctQ.

Using FRAP, we analyzed the exchange of EGFP-SctQ in strains expressing either SctD(WT) or SctD(2C), all expressed from their native sites on the pYV plasmid. In reducing conditions, where

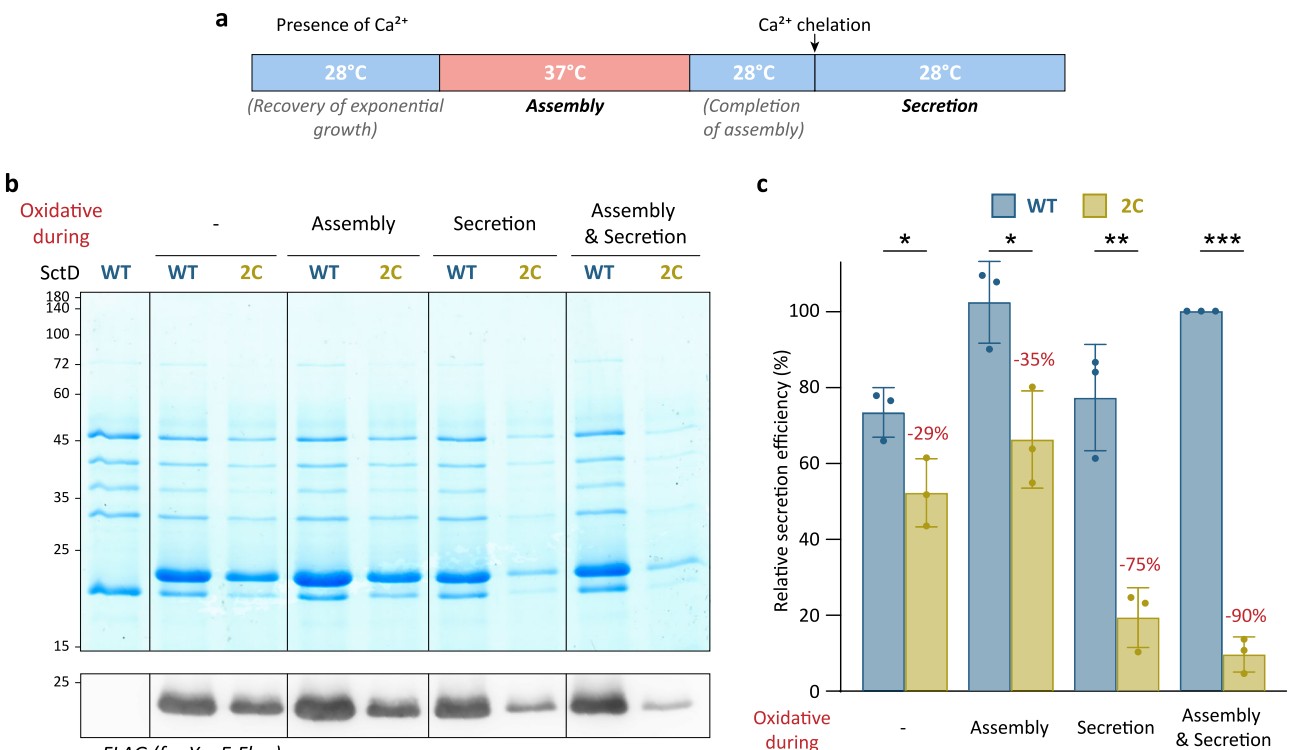

**Fig. 7 | Decreased SctD exchange slightly reduces effector export during assembly and even more strongly during secretion. a** Schematic timeline of the experiment displaying the assembly and secretion periods. **b** Representative secretion assay showing proteins in the supernatant corresponding to $3 \times 10^8$ bacteria. Bottom, Western blot anti-FLAG for quantification of YopE-FLAG export. $n = 3–4$ independent experiments. Expression of YopE-FLAG from the plasmid was induced using 0.2% L-arabinose. Except for the time spans indicated as "oxidative", crosslinking was prevented by the presence of 2 mM DTT. Left, molecular weight in kDa. **c** Quantification of relative secretion efficiency of YopE-FLAG in the strains shown in (**b**) ($n = 3$). Relative secretion efficiency was determined by Western blot quantification of the YopE-FLAG bands, compared with the strain expressing SctD(WT) under constant native oxidative conditions. Reduction of YopE-FLAG secretion in the SctD(2C) strain is indicated for each condition. Circles indicate single data points, bars show mean values, and error bars depict standard deviation. *, $p < 0.05$ with $p = 0.0291$ (SctD(WT) "-" vs SctD(2C) "-") and $p = 0.0201$ (SctD(WT) "assembly" vs SctD(2C) "assembly"); **, $p = 0.0033$; ***, $p = 4.5 \times 10^{-6}$ in a two-sided homoscedastic $t$ test.

SctD(2C) and SctD(WT) show similar exchange, fluorescence recovery of EGFP-SctQ was comparable between the two strains, with average half-time recovery ($t_{1/2}$) of $74.0 \pm 3.9$ s for SctD(WT) and $82.9 \pm 8.2$ s for SctD(2C) (Fig. 8a, Supplementary Fig. 11). Upon incubation with the chemical crosslinker BM(PEG)$_2$, which reduces exchange of SctD(2C), fluorescence recovery of EGFP-SctQ was slightly faster with an half-time recovery ($t_{1/2}$) decreased from $82.9 \pm 8.2$ s to $59.6 \pm 6.0$ s. While these results indicate that SctD exchange is negatively correlated with the exchange of the cytosolic T3SS components, the effect is mild and does not explain the positive correlation between SctD exchange and protein secretion by the T3SS.

### Exchange of SctD is required for the integration of the export apparatus into assembled injectisome membrane ring structures

During the assembly of the injectisome, the membrane-spanning rings and the export apparatus are the first structures to assemble. These substructures form in parallel and independently, as the absence of one does not entirely prevent the assembly of the other[18,23,24]. However, for the injectisome to be functional, the export apparatus must be integrated into the membrane-spanning rings and enclosed by the inner membrane ring formed by SctD and SctJ. This raises the question of how the large inner membrane complex of the export apparatus can be integrated if the membrane-spanning rings completely assemble beforehand. We wondered whether the exchange of SctD could facilitate the integration of the export apparatus into this substructure during injectisome assembly. To test this hypothesis, strains lacking the gene of the major export apparatus component *sctV* or the core

export apparatus component *sctS* were incubated at 37 °C in non-secreting medium to allow for the formation of the membrane-spanning rings in the absence of complete export apparatuses. After completion of the assembly, the expression of new T3SS components was repressed at 28 °C, and the expression of the missing export apparatus component (SctV or SctS) in trans was induced. To test if the export apparatus was assembled and integrated into the membrane-spanning ring complex, the bacteria were then transferred to secreting medium and secretion was quantified. The strain expressing SctD(WT) secreted proteins under these conditions, albeit at a reduced rate, most likely due to the time needed to express the complementing protein and integrate the substructure (see Supplementary Fig. 3 for experimental setup). This indicates that the export apparatus can integrate into assembled membrane ring structures if SctD is mobile (Fig. 9). In contrast, the strain expressing SctD(2C) only secreted proteins in reducing conditions suppressing the crosslinking. In oxidative conditions, where SctD(2C) exchange is strongly diminished, no secretion was observed, indicating that the export apparatus could not integrate, although the respective proteins were expressed (Supplementary Fig. 12). Altogether, these data are in line with the hypothesis that the exchange of SctD allows for the integration of the complete export apparatus into assembled inner membrane rings.

### Discussion

The type III secretion system is one of the most sophisticated bacterial nanomachines, whose assembly and function are tightly regulated[1,7,45]. At its core, a series of membrane rings encloses an inner membrane-spanning export apparatus[15,46]. The membrane rings consist of a highly

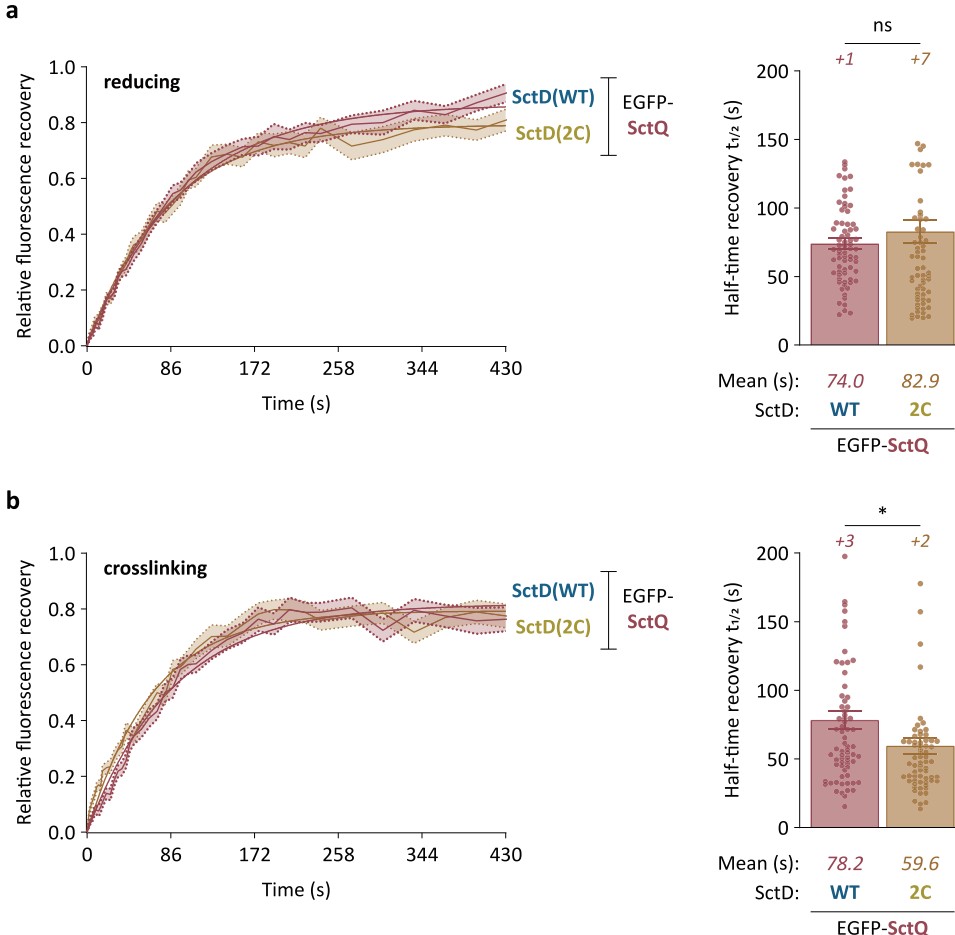

**Fig. 8 | The exchange of the cytosolic component SctQ does not require the exchange of SctD.** Left, fluorescence recovery curves of EGFP-SctQ representing the merged data from multiple foci for the indicated strains in reducing (**a**) and crosslinking (**b**) conditions environment (presence of 2 mM DTT and 0.1 mM BM(PEG)$_2$ during assembly and secretion, respectively). Average fluorescence recovery curves (light connecting lines) are depicted with their standard errors of the mean (shaded areas) and exponential regression curves (thick lines). ox.: native oxidative periplasmic environment; crossl.: irreversible crosslinking with chemical crosslinker (0.1 mM BM(PEG)$_2$). $n = 74$ for SctD(WT) red., 75 for SctD(2C) red., 81 for SctD(WT) crossl., and 80 for SctD(2C) crossl., from 3 independent experiments. Right, overall analysis of the half-time recoveries of the foci. The half-time recovery data for each strain were extracted from all individual fluorescence recovery curves with an $R^2$ value above 0.4 for exponential regression. $n = 71$ for SctD(WT) red., 67 for SctD(2C) red., 71 for SctD(WT) crossl., and 67 for SctD(2C) crossl. +$n$, additional values outside displayed y range (>200 s). Error bars represent the standard error of the mean. ns, $p = 0.3204$; *, $p = 0.0375$ in a two-sided homoscedastic $t$ test.

stable 15-mer ring structure in the OM formed by the secretin SctC, which is connected to the bitopic IM protein SctD, forming a 24-mer ring in the IM[15,16,29,40,47]. Although SctC and SctD interact via a hydrophobic interface[15], additional stabilizing interactions between neighboring SctD molecules are required for a functional interaction[47], indicating that it can be overcome in physiological conditions. Within the SctD ring, a smaller periplasmic 24-mer ring formed by the lipoprotein SctJ likely interacts with the export apparatus, whose SctR$_5$S$_4$T part moves from the IM towards the periplasm and interacts with SctU and a SctV nonamer[2,20,21,48]. During assembly of the *Y. enterocolitica* injectisome, the membrane rings and the export apparatus can form independently, with SctJ possibly interacting with the export apparatus before becoming part of the membrane rings structure[24]. The ability of the SctCD complex to form even in the absence of SctJ and the export apparatus[18] raises the question of how the export apparatus, comprising more than 70 IM transmembrane helices, and potentially together with SctJ, can be integrated into the fully assembled SctD ring structure. The finding that SctD proteins exchange between the injectisome-bound form and a pool in the IM may present an answer to this question. We first observed this exchange of SctD using functional assays, which showed a strong reduction of secretion upon

expression of a dominant negative SctD variant expressed after the assembly of functional injectisomes (Fig. 1b, c). Fluorescence microscopy provided complementary evidence, showing decreased fluorescence of assembled injectisomes, including natively expressed EGFP-SctD, upon the integration of unlabeled SctD monomers expressed after the completion of assembly (Fig. 2a, b). Next, SctD exchange was directly confirmed and quantified by fluorescence recovery after photobleaching. This benchmark method for investigating exchange of proteins in biological structures allows detection and quantification of the replacement of fluorescently labeled proteins that are photobleached in a diffraction-limited part of a cell, e.g,. the injectisome clusters in *Y. enterocolitica*[18,49], by non-bleached proteins from other parts of the bacterium. SctD is replaced with a half-time of slightly below three minutes (Fig. 3), slower than the cytosolic component SctQ, but much faster than what could be explained by formation of new injectisomes by newly synthesized proteins (Supplementary Fig. 5)[29,31].

Given the fact that SctD can be purified as part of SctCDJ "basal body" structures[50–53] and "needle complexes" containing SctCDJ and the needle[47,51,52], and in situ structural analyses do not provide evidence for a significant subset of incomplete SctD rings[11–13], an exchange of

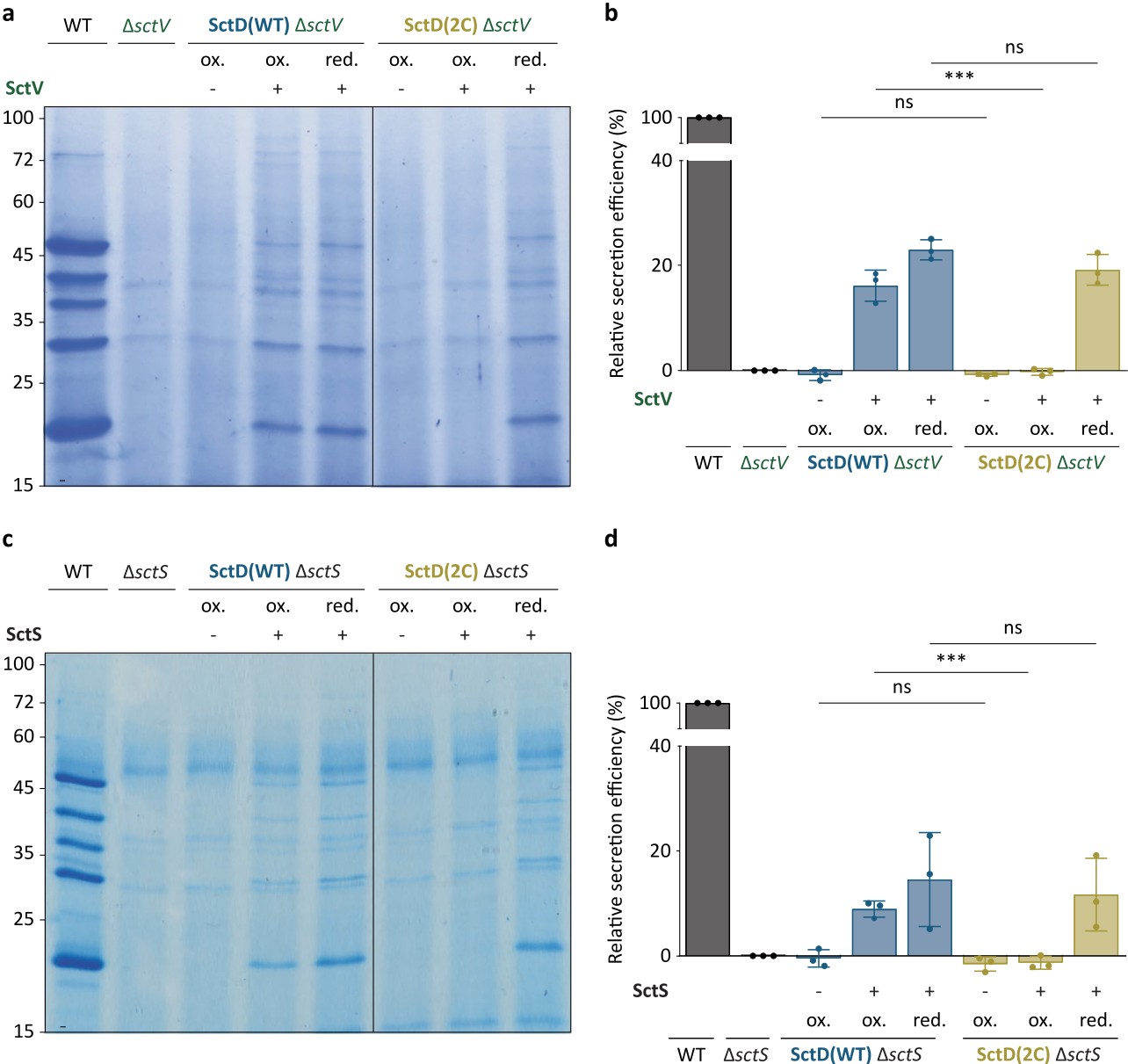

**Fig. 9 | The exchange of SctD is important for the integration of the export apparatus into assembled inner membrane ring structures, allowing for the assembly of the T3SS.** Secretion assays assessing the integration of the export apparatus components SctV (**a**) and SctS (**c**) after assembly of the injectisome basal body (n = 3). Normalized amounts of supernatant corresponding to $6 \times 10^8$ bacteria per lane were loaded on an SDS-PAGE gel and stained with Coomassie Brilliant Blue. Left, molecular weight in kDa. ox.: oxidative periplasmic environment; red.; reductive periplasmic environment (presence of 2 mM DTT); −/+ indicate no expression and expression by addition of 0.2% arabinose, respectively, of SctV (**a**) or SctS (**c**). **b**, **d** Quantification of relative secretion efficiency of the effector YopE

(band below 25 kDa) in (**a**, **c**). Relative secretion efficiency was determined by densitometric analysis of SDS-PAGE gels after staining for the YopE band, compared to the average secretion in the wild-type strain for each condition. Bars show the mean values, while error bars depict the standard deviation. Circles indicate single data points. **b** ns, $p = 0.9280$ (SctD(WT) $\Delta sctV$ -SctV "ox." vs SctD(2C) $\Delta sctV$ -SctV "ox.") and $p = 0.1328$ (SctD(WT) $\Delta sctV$ +SctV "red." vs SctD(2C) $\Delta sctV$ +SctV "red."); ***, $p = 0.0007$; (**d**) ns, $p = 0.4398$ (SctD(WT) $\Delta sctS$ + SctS "ox." vs SctD(2C) $\Delta sctS$ + SctS "ox.") and $p = 0.6838$ (SctD(WT) $\Delta sctS$ + SctS "red." vs SctD(2C) $\Delta sctS$ + SctS "red."); ***, $p = 0.0009$ in a two-sided homoscedastic $t$ test.

SctD in secreting bacteria may be counterintuitive at first glance, especially given that SctD is a central part of the core membrane ring structure formed by SctCDJ. However, subunit exchange is a common biological phenomenon[37] and similar observations were made for other biological structures, including the C-ring of the flagellum[54–56] and the nuclear pore complex[57,58]. One possible explanation for this discrepancy is that the extraction and purification procedures for the structural analyses promote the stability of these complexes. An even simpler possibility is that SctD dissociation requires the presence and concurrent integration of a replacing SctD subunit. This would mean that at any given time, the majority of the 24 SctD sites per injectisome

are occupied, providing a stable link between SctC and SctJ at neutral pH. In line with this interpretation, a quantification of fluorescence intensities of foci for labeled T3SS components in live *Y. enterocolitica* showed highly similar distributions for SctD, -K, -Q, -L, and -N[30], indicating no or very few incomplete injectisomes at any given time, which in turn is supported by single molecule quantification for *Salmonella* SPI-1[59] and in situ structural studies for various species[11–13,40,60–62].

What are the exchange units of SctD? In theory, FRAP should allow to count the recovery steps, with 24/n steps indicating the exchange of SctD n-mers. However, while a similar method, stepwise photobleaching, has been previously used to determine complex

stoichiometry in ideal cases[63,64], the fact that injectisomes cluster in diffraction-limited foci[11,49] and the noise in the fluorescence measurements complicates this analysis. Interestingly, the analysis of reversible oxidative SctD(2C) crosslinking and irreversible chemical crosslinking by BM(PEG)$_2$ revealed two distinct higher-molecular weight fractions, one corresponding to SctD trimers or tetramers, and an even larger one (Fig. 4e). Both the formation of more closely associated trimers and tetramers within the SctD ring are conceivable: While the SctC:D interface in the periplasm has a 2:3 ratio due to the integration of a 16$^{th}$ SctC domain at the interface with the SctD 24-mer[15], there is a 4:1 stoichiometry of SctD: K in the cytosol[12,65]. Given that we detected neither a clear stepwise recovery of fluorescence, nor diffusing clusters of non-injectisome-bound EGFP-SctD fluorescence, indicative of unbound SctD oligomers, in micrographs, the exchange of SctD monomers between a pool in the IM and the injectisome is the most likely scenario in our view. However, the exchange of SctD oligomers, e.g., trimers or tetramers, cannot be excluded from our data.

While protein exchange in biological complexes is a frequent phenomenon, its functional significance has been determined in a few cases so far[37]. For the T3SS, a clear correlation between protein exchange of the cytosolic subunit SctQ and secretion has been observed[29]. Furthermore, the presence of non-injectisome-bound cytosolic SctQ is essential for secretion, as shown by the fact that optogenetic sequestration of SctQ reversibly inhibits secretion[32]. However, demonstration of a direct causal link between the dynamics and function of the T3SS requires the ability to control protein exchange. The well-described structure of SctD in its native context[15,38,39,50] allowed us to perform this key step in the present study. Based on a structural homology model and molecular dynamics simulation, we identified a pair of amino acids, R161 and E179, which are in close contact in neighboring SctD subunits in the assembled injectisome (Fig. 4). These residues were mutated to cysteines, generating the SctD(2C) variant that enables targeted crosslinking. Indeed, in strains expressing the SctD(2C) variant, protein exchange can be reversibly inhibited by the formation of disulfide bonds in the oxidative periplasmic environment of Gram-negative bacteria, or irreversibly by chemical crosslinking (Fig. 5). Strikingly, reducing SctD exchange also reduces type III secretion (Fig. 6), indicating a direct causative relation. It should be noted that other, more indirect effects of SctD crosslinking are conceivable, such as the inhibition of signaling pathways for the activation of the injectisome as described for *Shigella*[66–68] and *Salmonella*[69]. However, the pathway for this regulatory mechanism, which has not been demonstrated in *Yersinia*, needle tip – needle – export apparatus, does not involve SctD and the probability that crosslinking of the two selectively introduced cysteine residues would specifically perturb such a pathway, both through the native reversible disulfide bond formation and through the relatively long and flexible crosslinker BM(PEG)$_2$ – is low. But which step of the secretion process requires SctD exchange? Our data support the hypothesis that SctD exchange is required for the integration of the IM export apparatus into pre-assembled membrane-spanning ring structures (Fig. 9). This is relevant under native conditions, albeit to a lower level (Fig. 7), probably because some SctCD rings can form around existing export apparatuses. Besides the integration of the export apparatus, SctD exchange might also be relevant for its release, either to replace or to disassemble parts of the injectisome, as was observed in response to low external pH[34].

In addition to the role of SctD exchange during the assembly of the injectisomes, we also noted a significant reduction of protein secretion when SctD exchange was specifically inhibited during secretion in bacteria that do not assemble new injectisomes. While this effect cannot be fully explained at the moment, we found that it is not conveyed by a requirement of SctD exchange for the dynamics of SctQ, which is a part of shuttle structures recruiting effectors in the cytosol[31] (Fig. 8). Interestingly, SctD has no homolog in the flagellum and is one of the least conserved proteins in the T3SS with a far lower sequence conservation than its interaction partners SctC and SctJ[17,70]. This may allow both for structural adaptation, e.g. to accommodate different distances of the bacterial membranes[40], and for species-specific functional adaptation, e.g. to convey the response to external pH in gastrointestinal pathogens[34], and may also influence SctD exchange. It would therefore be promising to study the effect in other bacteria and in different relevant conditions in future experiments.

To this date, protein exchange in functional complexes has mainly been shown for soluble proteins[37]. A prominent example of membrane proteins is the load-dependent association of MotA$_5$B$_2$ stator complexes to the flagellar C-ring[71,72]. However, these stator units are distinct structural entities, whereas SctD is more intricately integrated into the injectisome structure. The exchange of a core membrane component of a biological complex shown in this study might therefore serve as a paradigm for other proteins, and the approach used in this study could be used for their analysis. Ring-building motifs found in SctD, composed of two alpha-helices folded against a beta-sheet[38], are also present in unrelated proteins, such as SpoIIIA proteins, which participate in a channel between mother cell and forespore during sporulation of *Bacillus subtilis*[73,74]. Future analysis of these biological complexes will reveal if the exchange of core membrane subunits, shown here for the T3SS component SctD, is a highly specific adaptation for the type III secretion system or a common principle in microbiology.

## Methods

### Nomenclature
While a core of T3SS proteins is shared between the injectisome and the evolutionarily related bacterial flagellum[75–77], this manuscript uses T3SS to refer to the injectisome, and applies the common T3SS protein nomenclature[8,78].

### Bacterial strain generation and genetic constructs
A list of the strains and plasmids used in this study is provided in Supplementary Table 1, oligonucleotides used are detailed in Supplementary Table 2. All *Y. enterocolitica* strains used in this study are based on the wild-type strain MRS40 or its derivative IML421*asd* (ΔHOPEMT*asd*). IML421*asd* is genetically engineered to lack all major virulence effector proteins (YopH, -O, -P, -E, -M, -T), and carries a deletion in the aspartate-beta-semialdehyde dehydrogenase (*asd*) gene, making it auxotrophic for diaminopimelic acid (DAP), and thus suitable for work within a Biosafety Class 1 environment[40]. For ectopic protein expression in *Y. enterocolitica*, the corresponding genes were cloned into pBAD-His/B-based expression plasmids (Invitrogen). These plasmid constructs were sequence-verified before being introduced into the respective bacterial strain via electroporation. Unless specified otherwise, all other fusion proteins in this study were expressed as endogenous translational fusions, introduced into the native genetic background through allelic exchange[79].

### Bacterial cultivation
Overnight cultures of *Y. enterocolitica* were grown at 28 °C in a shaking incubator using Brain Heart Infusion (BHI) medium supplemented with nalidixic acid (Nal, 35 μg/ml) and, if necessary, diaminopimelic acid (Dap, 60 μg/ml). For strains carrying pBAD-based plasmids, ampicillin (Amp, 200 μg/ml) was added to ensure plasmid stability. To prepare day cultures, overnight stationary-phase cultures were inoculated into fresh BHI medium supplemented with 35 μg/ml Nal, 60 μg/ml Dap (if required), 20 mM MgCl$_2$, 0.4% glycerol, 200 μg/ml Amp (if required), and either 5 mM ethylene glycol-bis(β-aminoethyl ether)-N,N,N′,N′-tetraacetic acid (EGTA, to induce T3SS secretion) or 5 mM CaCl$_2$ (to inhibit T3SS secretion) to an OD$_{600}$ of 0.15 (secreting conditions) or 0.12 (non-secreting conditions), and then incubated for 90 min at 28 °C. To induce the expression of the *yop* regulon from its native locus

on the pYV plasmid, the cultures were rapidly shifted to 37 °C and kept at this temperature for 120 min, unless specified differently, to allow for injectisome assembly.

## Exchange of SctD variants expressed from the native locus by a dominant negative mutant

After initial growth under non-secreting conditions as described in the "Bacterial cultivation" section, the bacteria were shifted back to 28 °C for 120 min to repress further expression of injectisome components and stop the assembly of new injectisomes. At the time of this temperature shift, expression of wild-type SctD or dominant negative mutant SctD(L4) from pBAD-His/B derivative plasmids was induced by the addition of 0.01% L-arabinose or repressed by 0.2% L-glucose. Following this incubation, bacteria were pelleted by centrifugation (6 min at 4000 g, room temperature (RT)), washed with 1 ml filtered phosphate-buffered saline (PBS, 8 mg/mL NaCl, 0.2 mg/mL KCl, 1.78 mg/mL Na$_2$HPO$_4$ ·2H$_2$O, 0.24 mg/mL KH$_2$PO$_4$, pH 7.4), and pelleted again. The bacteria were then resuspended in secretion-inducing medium (BHI medium supplemented with 35 μg/ml Nal, 20 mM MgCl$_2$, 0.4% glycerol, 200 μg/ml Amp, 5 mM EGTA, and 0.01% L-arabinose or 0.2% L-glucose) and incubated at 28 °C for 60 min to trigger the secretion of native injectisome substrates into the supernatant.

## Crosslinking assays

The initial incubation at 37 °C, described in the "Bacterial cultivation" section, was performed in the presence of 0.03% L-arabinose to induce expression of the SctD variants during injectisome assembly, and 2 mM DTT to prevent the natural disulfide bond formation. Following this, the bacteria were pelleted (6 min at 4,000 g, RT), washed with 1 ml filtered PBS, and pelleted again. The washed pellets were resuspended in fresh secretion-inducing conditions (BHI supplemented with 35 μg/ml Nal, 20 mM MgCl$_2$, 0.4% glycerol, 200 μg/ml Amp, 5 mM EGTA, and 0.2% L-glucose) with 2 mM DTT (reductive periplasmic conditions), or treated with 0.1 mM BM(PEG)$_2$ (to induce irreversible crosslinking in natural oxidative periplasmic conditions) as needed, and incubated at 28 °C for 180 min. This incubation repressed further injectisome component expression and stopped the assembly of new injectisomes, while allowing for the secretion of the native injectisome substrates into the supernatant.

## Quantification of secretion efficiency of YopE-FLAG

The initial incubation at 37 °C, described in the "Bacterial cultivation" section, was performed in the presence of 0.2% L-arabinose to induce expression of YopE-FLAG from pBAD during injectisome assembly and, where indicated, 2 mM DTT for reducing conditions. Subsequently, bacteria were shifted back to 28 °C for 60 min to repress further expression of injectisome components and stop the assembly of new injectisomes. After this incubation, bacteria were pelleted (6 min at 4000 g, RT), washed with 1 ml filtered PBS, and pelleted again. They were then resuspended in fresh secretion-inducing medium (BHI supplemented with 35 μg/ml Nal, 20 mM MgCl$_2$, 0.4% glycerol, 200 μg/ml Amp, 5 mM EGTA, 0.2% L-arabinose, and, where indicated, 2 mM DTT to prevent the potential formation of natural crosslinking (disulfide bonds) during injectisome secretion) and incubated at 28 °C for 180 min. This incubation repressed further injectisome component expression, stopped new injectisome assembly, and initiated secretion of native injectisome substrates and YopE-FLAG.

## Integration of the export apparatus after membrane ring assembly

The initial incubation at 37 °C, described in the "Bacterial cultivation" section, was performed in the presence of 0.2% L-glucose to prevent expression of SctV or SctV-FLAG from the pBAD-based plasmids during injectisome assembly. Subsequently, bacteria were shifted back to 28 °C for 60 min to repress further expression of injectisome components and stop the assembly of new injectisomes. After this incubation, bacteria were pelleted (6 min at 4000 g, RT), washed with 1 ml filtered PBS, and pelleted again. They were then resuspended in fresh secretion-inducing conditions (BHI supplemented with 35 μg/ml Nal, 20 mM MgCl$_2$, 0.4% glycerol, 200 μg/ml Amp, 5 mM EGTA, and, where indicated, 2 mM DTT to prevent the potential formation of natural crosslinking (disulfide bonds) during injectisome secretion) and incubated at 28 °C for 180 min. This incubation repressed further injectisome component expression, stopped new injectisome assembly, and initiated secretion of native injectisome substrates. At this point, 0.2% L-arabinose or 0.2% L-glucose was added to control the expression of SctV or SctV-FLAG from the pBAD-based plasmids.

## Secretion assays, and total cell protein assays

For secretion assays, 2 ml of culture was centrifuged (10 min at 21,000 g, 4 °C) and 1.8 ml of the culture supernatant was carefully collected, while the total cell pellet was kept for further analysis. Proteins in the supernatant were precipitated with 10% trichloroacetic acid (TCA) at 4 °C for 1–16 h and recovered by centrifugation (15 min at 21,000 g, 4 °C). The resulting precipitated protein pellet was then washed with 1 ml ice-cold acetone and resuspended in SDS-PAGE loading buffer (1% sodium dodecyl sulfate, 5 mM Tris, 10% glycerol, 50 mM DTT, where indicated, bromophenol blue, pH 6.8) in a volume normalizing to 0.6 OD units (ODu) for secretion assays and 0.3 ODu for total cell analysis per 15 μl, unless indicated otherwise (1 ODu = 1 ml of culture at OD$_{600}$ of 1, ~ 5 × 108 Y. enterocolitica). The samples were heated for 10 minutes at 99 °C. Protein separation was performed by loading 15 μl samples on 15% SDS-PAGE gels or 4–20% Mini-Protean® TGX™ precast protein SDS-PAGE gels (Bio-Rad; "Crosslinking assays" section), and protein sizes were determined using the BlueClassic Prestained Marker (Jena Biosciences) as a standard. The gels were stained with FastGene-Q-stain (NipponGenetics) for visualization. For immunoblotting, the separated proteins were transferred onto a nitrocellulose membrane using a semi-dry transfer system (1.3 A; 25 V; 7 min for standard blots or 1.0 A; 25 V; 1 hour for crosslinked complex blots). Primary rabbit anti-SctD antibody (MIPA232, 1:1,000) or primary rabbit anti-FLAG antibody (Rockland 600-401-383, 1:2,000) was used, followed by a secondary anti-rabbit antibody conjugated to horseradish peroxidase (Sigma-Aldrich A8275 1:10,000). Detection was performed using Immobilon Forte chemiluminescence substrate (Sigma-Aldrich) on a LAS-4000 Luminescence Image Analyzer. Band intensity in both stained SDS-PAGE gel and immunoblot was determined by gel densitometry using the Fiji software (ImageJ 1.51 f/1.52i/1.52n, https://fiji.sc/)[80]. The data were plotted using GraphPad Prism version 10.1.0 for Windows (GraphPad Software, Boston, Massachusetts, USA). For normalization, the strain expressing SctD(WT) and incubated under oxidative conditions was used as a 100% reference for the other strains and conditions.

## Molecular modeling methods

We simulated the Y. enterocolitica SctC, SctD (using only the periplasmic domain from Gln158–Ala383), and SctJ complex model (herein YeSctCDJ) at pH 7.4 and analyzed the most frequent interactions within the protein-protein interface. The YeSctCDJ model was generated based on the structure of the Salmonella SPI-1 injectisome (PDB ID 6PEM)[15] with a defined SctC:D: J interface. The homology model was executed with Prime 2021.3[81] (Schrödinger LLC), which was followed by Protein Preparation Wizard 2021.3[82] (Schrödinger LLC).

We used the Desmond MD simulation engine[83] and the OPLS3e force-field[84]. The prepared systems were solvated in a cubic box with the size of the box set as 13 Å minimum distance from the box edges to any atom of the protein, with periodic bound conditions. TIP3P water model[85] was used to describe the solvent, and the net charge was neutralized using Na+ ion with a final salt concentration of 150 mM.

RESPA integrator timesteps of 2 fs for bonded and near and 6 fs for far were applied. The short-range coulombic interactions were treated using a cut-off value of 9.0 Å, whereas long-range coulombic interactions were estimated using the Smooth Particle Mesh Ewald method[86]. Before the production simulations, the systems were relaxed using the default Desmond relaxation protocol. Simulations were run in an NPT ensemble, with a temperature of 310 K (using the Nosé-Hoover thermostat)[87] and pressure of 1.01325 bar (Martyna-Tobias-Klein barostat)[88]. For each system, a combination of three independent simulations of 500 ns was carried out, resulting in at least -1.5 μs simulation data for each system. Maestro simulation interaction analysis tool (Schrödinger LLC) was used for the analysis of RMSD and interaction analysis. All raw data trajectories, relevant conformations, and interaction data are available at https://doi.org/10.5281/zenodo.13134138.

## Fluorescence microscopy
For fluorescence microscopy, bacteria were cultivated according to the specific experimental conditions. 800 μl of culture was collected by centrifugation (4 min at 2400 g, RT) and resuspended in 400 μl of minimal medium [100 mM 2-[4-(2-Hydroxyethyl)piperazin-1-yl]ethane-1-sulfonic acid (HEPES) (pH 7.2), 5 mM $(NH_4)_2SO_4$, 100 mM NaCl, 20 mM sodium glutamate, 10 mM $MgCl_2$, 5 mM $K_2SO_4$, 50 mM 2-(N-morpholino) ethane sulfonic acid (MES), and 50 mM glycine supplemented with 0.2% casamino acids, 60 μg/ml Dap (if required), either 5 mM EGTA (secreting conditions) or 5 mM $CaCl_2$ (non-secreting conditions)]. 2 mM DTT or 0.1 mm BM(PEG)$_2$ were added as indicated. From this bacterial suspension, 2 μl were spotted onto an agarose pad (1.5% low melting agarose (Sigma-Aldrich) in minimal medium with 0.2% casamino acids, 60 μg/ml Dap (if required), and either 5 mM EGTA (secreting conditions) or 5 mM $CaCl_2$ (non-secreting conditions), plus 2 mM DTT or 0.1 mM BM(PEG)$_2$ where indicated) on glass depression slides (Marienfeld). Imaging was conducted using a Deltavision Elite Optical Sectioning Microscope equipped with a UPlanSApo 100×/1.40 oil objective (Olympus) and an EDGE sCMOS 5.5 camera (Photometrics). The GFP signal was captured using a GFP filter set (excitation: 480/25 nm, emission: 535/28 nm) with an exposure time of 0.2 s for FRAP and up to 0.8 s for standard fluorescence microscopy, acquiring z-stacks with 9 slices (Δz = 0.15 μm) per fluorescence channel. Image processing and fluorescence quantification were conducted using the Fiji software (ImageJ 1.51 f/1.52i/1.52n)[80]. For visualization purposes, selected fields of view were identically adjusted for brightness and contrast across the compared image sets.

## Quantification of fluorescent foci
To quantify the number of stable EGFP-SctD and EGFP-SctQ foci, individual *Y. enterocolitica* cells were segmented using the StarDist neural network. The training dataset was generated by manually annotating 17 fields of view of *Y. enterocolitica* cells grown in BHI medium and imaged on agarose pads on a Deltavision Elite Optical Sectioning Microscope. To account for variations in the imaging plane, z-stacks with 150 nm spacing were recorded, and 5 representative slices were selected for network training. This resulted in 85 images, which were quartered, producing a final dataset of 300 (training) and 40 (validation) image pairs. During training, the dataset size was increased 4-fold through flipping and rotation. The StarDist model was trained for 100 epochs on the ZeroCostDL4Mic platform, using a batch size of 4, 260 steps, 120 rays, a grid size of 2, an initial learning rate of 0.0003, and a train/test split of 80%/20%. Both the dataset and the model are available in the DeepBacs collection on Zenodo (https://doi.org/10.5281/zenodo.11105050 [89]). Foci localization was performed using the ThunderSTORM plugin in the Fiji software[80]. Localizations were identified using 3rd-order B-Spline filtering and local maximum approximation, with peak intensity thresholds set at 2-fold (EGFP-SctD) to 4-fold (EGFP-SctQ) the standard deviation of the first wavelet

level. For sub-pixel localization, peaks were fitted with an integrated Gaussian function within a 3-pixel fitting radius. Localizations were then filtered by PSF width (sigma <600 nm) and rendered as a scatter plot with a magnification of 1. Foci counting was achieved by measuring the integrated intensity within the ROIs of segmented cells. To ensure accurate foci localization within the cell boundaries, each ROI was expanded by 2 pixels using a custom-written Fiji macro[90].

## Fluorescence recovery after photobleaching (FRAP)
Following an extended incubation at 37 °C for 180 min under secreting conditions as detailed in the "Bacterial cultivation" section in the presence of 2 mM DTT or 0.1 mM BM(PEG)$_2$ in the day culture as indicated, cells were prepared as outlined in the "Fluorescence Microscopy" section. To minimize photobleaching, no z-stacks were acquired. Three pre-bleach images were captured, and specific locations near selected bacterial cell poles (regions of interest (ROI) including individual fluorescent foci) were photobleached using individual 30 ns pulses of a 488 nm laser. The recovery of the bleached foci was monitored via time-lapse microscopy over 8 minutes.

The relative fluorescence of the bleached spot was calculated using the formula:

$$Relative\,fluorescence = \frac{Intensity_{ROI} - Intensity_{background}}{Intensity_{total\,cell} - Intensity_{background}} \quad (1)$$

This relative fluorescence was normalized, with the average pre-bleach value set to 1, and the post-bleach value set to 0. Exponential recovery was fitted using the following equation:

$$F(t) = A^* \left(1 - e^{-k^*t}\right) \quad (2)$$

where $F(t)$ is the relative fluorescence intensity at time $t$ after photobleaching, $A$ represents the overall ratio of fluorescence recovery, and $k$ is the rate constant of the recovery process. For the fit, $A$ was constrained to values between 0.7 and 1.1. Each bleach curve was fitted individually, and the resulting data were merged to obtain average fluorescence recovery curves for each strain. Fitted curves with $R^2 > 0.4$ were included in the analysis to measure the half-time of recovery ($t_{1/2}$), which was calculated from the time constant using the following equation:

$$t_{1/2} = \frac{\ln(2)}{k} \quad (3)$$

For the representative images, photobleaching over the time-lapse was corrected using the Bleach Correction plugin in the Fiji software[91]. The "Simple Ratio" method was applied, using the average fluorescence intensity of the background in the first time-lapse image. Selected fields of view were uniformly adjusted for brightness and contrast across image sets by applying the "Enhance Contrast" function with 0.35% saturated pixels.

## Statistical analysis
To evaluate the statistical significance of quantitative differences, unpaired two-tailed t tests were used, unless specified differently.

## Reporting summary
Further information on research design is available in the Nature Portfolio Reporting Summary linked to this article.

# Data availability
Raw data trajectories, relevant conformations and interaction data of the molecular modeling of SctCDJ are available on Zenodo (https://doi.org/10.5281/zenodo.13134138). The dataset and the model used for the quantification of the foci using DeepBacs are available in the DeepBacs

collection on Zenodo (https://doi.org/10.5281/zenodo.11105050). All other relevant data are included in the paper and/or its supplementary information files. Source data are provided with this paper.

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

## Acknowledgements

Work in the Diepold group was supported by the Max Planck Society. T. Kronenberger receives funding from the German Center for Infection Research (DZIF, TTU06.716). The authors would like to thank Christoph Spahn, Max Planck Institute for Terrestrial Microbiology, Marburg, and University of Würzburg, for support with machine learning and optimizing the quantification of the fluorescent foci, and the CSC-Finland for generous computational resources.

## Author contributions

C.B. performed the majority of measurements and data processing and participated in writing and revising the manuscript. S.W. performed initial experiments that defined the project. T.K. contributed the molecular modeling measurements and figures. A.D. provided the study concept, supervision, and participated in data analysis and writing, and revision of the manuscript.

## Funding

## Competing interests

The authors declare no competing interests.
