## [Transparent Peer Review file · Nature Communications]

Continuous exchange of an inner-membrane ring component is required for assembly and function of the type III secretion system

Corresponding Author: Professor Andreas Diepold

Version 0:

Reviewer comments:

Reviewer #1

(Remarks to the Author)

Manuscript ID: NCOMMS-24-57188

Title: Continuous exchange of the inner membrane ring component SctD is required for the assembly and function of the type III secretion system

General comment: The authors have performed a wonderful study regarding the membrane ring component SctD and have proven its continuous exchange and mobility at the inner membrane, a process that seems to be important or even essential for protein secretion through T3SS in *Yersinia*.

The study provides a novel insight into the mechanistic model of injectisome assembly and the experiments performed are a major breakthrough in the field. The secretion process manipulation, the reversible cross-linking and the FRAP experiments presented in the paper provide a step-to-step dissection of the system in vivo that appears to be a powerful tool/pipeline to study bacterial secretion.

The manuscript is well written with a nice flow, experimental procedures seem to be clear and the results robust and conclusive. However, some points the authors made need to be clarified better and/ or described in more detail, so that the reader is not confused or even mis-led. Additionally, some control experiments would be essential in some cases so to strengthen their point.

Overall, according to my opinion, this manuscript should be accepted for publication, if the authors address the major and minor comments listed below.

Major comments:

1: My first major comment is about Figure 2 and Figure S2. Unfortunately, I found it very confusing and although I understand the purpose and its importance, I fail to follow it either in the text and or in the figure. According to my opinion, the authors should re-design Figure 2 and perhaps merge it with S2 as the data presented in S2 are essential to support the manuscript. Of course, the manuscript will need revision to match the new Figure 2.

As an example, in Figure 2A, the authors show one micrograph on the right and mention "wt". What is this wt stands for? No plasmid in those cells? It contains the plasmid but there has been no addition of arabinose? Has it grown in parallel, collected and analysed at the end of the experiment? A clear description is required. In Figure S2A similar question.

Additionally, in Figure S2, what do the micrographs represent? The (-) and (+) images both shown foci. Do they come from the same cultures as from Fig. 2A with no arabinose added and incubated for the same time? Or is the cultures before adding arabinose? If the latter case is correct, I found those images redundant since you have the "wt" micrograph and they are shown in Fig. 2. I suggest the authors to redesign the figures and write more descriptive legends.

2: There is some discrepancy regarding SctQ membrane localization and efficient exchange. When the authors used SctD (L2) mutant that has low/ no exchange, SctQ exchange is also affected. They explain this phenotype due to the lack of SctK interaction with SctD. Later in the text, when using SctD (2C) mutant that again shows no SctD membrane exchange, but SctQ one is not affected. This is because SctD is immobilized in the membrane and we make the assumption that SctK is not affected from that? The authors should elaborate all options, and discuss a little bit further those two different results. How those two different results are justified? Does SctQ need the SctD to be mobile or not?

3: Another section needs further clarification, the cross-linking experiments presented in Figure 4. As a start, it seems that the wt cysteines are present in the SctD (2C) protein. Is that correct? If so, it should be clearly mentioned in the text and explain why the native cysteines were retained. Additionally, one would expect that those cysteines should also interact and cross-link with something in close proximity, especially when non-specific BM(PEG)2 was used. How do the authors justify this result? Why those native cysteine are not reactive?

#4: Regarding the cross-linking experiments, one would also expect many more and of high-molecular weight cross-linking products, having in mind that SctD is forming 24mer ring-like structures at the membrane, interacts with SctK, SctJ, SctV and SctC rings. How the authors secure that the in vivo cross-linking results shown in Figure 4E are only due to SctD-SctD intra-molecular cross-linking? Why SctD could not interact with other components that they are in close proximity/interacting at the membrane? It is possible that those products are not easily analysed on gel, i.e due to size (too big to resolve in the gel). Do the authors observe protein precipitation in the well of the gels? If yes, this should be discussed.

MD simulations combined with analysis of structural models indeed can predict interaction sites and no one argues against the validity of those experiments. However, taking the current results and according to the protein size shown in the gel, someone can argue that SctJ-SctD and/or SctD-SctV complexes could be the cross-linked products detected. To strengthen their results, the authors should use an antibody against other injectisome proteins (i.e SctV, SctJ) in Figure 5E, so to verify no cross-linking between those components and SctD.

This is an important statement to be made firmly and clarified, since the rest of the paper relies on the fact that SctD protomers cannot be exchanged and this intra-protomeric dynamic exchange is essential for T3SS to function properly.

5: Finally, my main concern is about the final statement the authors make regarding the integration of the export apparatus and how this depends on SctD exchange. I think that the experiments performed are not sufficient to make such a firm statement. Perhaps it can be suggested or hypothesised but more experiments are needed in order to prove their point. In more detail, the in vivo secretion results in Figure 9 show hardly any complementation, if one is to compare the wt profile with the complemented strain (comparing lane 1 with lanes 4-5), suggesting that the system is compromised by those gene deletions and cannot be properly restored. I think the authors should make a comment on the text.

Moreover, if we focus and compare lanes 7 and 8, indeed there is a more prominent defect in secretion under oxidised conditions and this in vivo secretion defect is directly correlated by the authors with the lack of SctV integration at the export apparatus in the membrane. This is a statement that cannot be extrapolated firmly from in vivo secretion results. The authors present no physical evidence regarding SctV membrane integration, or even membrane localization and how it might be effected by SctD. SctV is the major component of the export apparatus and plays a major role in injectisome assembly, ATPase complex stabilization and secretion hierarchy. One could argue that the rest of the injectisome is affected or even downstream events are compromised in the absence of SctV per se.

I think the authors should perform a similar experiment using sctJ or sctC or sctU cells as a control to show that under those conditions the secretion is restored and cross-linked SctD has no defective role in secretion through the system. Additionally, a FRAP experiment like the one performed for SctQ (Figure 8) should be performed for SctV to show no recovery after photo-bleaching is detected when SctD (2C) is cross-linked. Those results would make their statement more robust.

Otherwise, the authors with the current state of the manuscript should lower the strength of their conclusions all across the text. i.e lane 381: "... SctD allows for the integration..." could be re-phrased like could allow, we speculate it allows. Same for lane 457: "...our data unambiguously..." I think that unambiguously is too strong for that statement and they should consider re-phrasing and/or deleting, and authors could explore other possible conclusions in the discussion.

Minor comments:

1, line 29: ...expressed at 37o... A "C" is missing

2, line 76: ...as previously thought. I think a reference is missing here

3, line 163: ...with kinetics slightly slower... According to my opinion 'slightly' should be removed because it gives the impression of fast exchange. Since SctD exchange is 3 times slower compared to SctQ and 2 times faster compared to SctV, one could argue and say that SctD exchange is slightly faster than SctV.

4, line 192: classical molecular dynamics... I think a reference is missing here

#5: line 207-217: Here the authors are describing the cross-linking experiment. I am wondering why there are so low signals of the monomeric SctD especially in the non-cross-linked lane of the SctD wt and 2C. Could it be that the protein is degraded or even aggregating and not resolved in the gel? Could that be an indication of high molecular species formed? I think the authors should elaborate more on this result and justify it better on the text.

6, line 581: using Fiji... the word software is missing here. Perhaps also the link of Fiji.

7, line 581: How data were plotted and normalized is not described. Percentage of the wt? What is considered wt? etc. I think this information should be also written in this section.

8: All across the text, the authors mention different bacterial growing conditions in which they manipulate the system and can have protein secretion or not, injectisome assembly or not, etc. For nonYersinia experts, it will be difficult to immediately connect the experimental conditions with the results shown, so to follow the flow of the text.

Could the authors consider adding a table listing the different conditions of bacterial growth (incubation time, temperature, medium composition) in supplement or in materials and methods and correlate it with T3SS step they monitor? That would provide some extra clarity to the text.

9: The authors should use the same font and size all across the Figures. For example: in Figure 6 the numbers in A are smaller than in B; in Figure 7 the same and in Figure 9 although the numbers are the same size all across the figure, they are smaller than the rest of the figures.

10: It would have been easier for the reader if western blots are directly associated with the protein tracked and the antibody used. In that case, one won't have to go through the text or Figure legend to find this information. i.e. In Fig. 4E add

anti-Flag- SctD and in 7B add anti-Flag-YopE.

11: In Figure 6B, the authors are comparing the secretion efficiency of sctD complemented with SctD (2C) under oxidative or reduced conditions. In panel B though they show only two bar graphs. Additionally, in the legend is mentioned that secretion from wt was considered in each condition as 100%. Why is that? Why not to keep “wt oxidized” as 100% (physiologically relevant condition) and compare every other condition with this?

I think that the authors should show in B all the bar graphs from all the conditions quantified from A. Moreover, they should use one condition as 100% and express all the other ones as a percentage of this.

12: Same comment as above for Figure 7B and C. I think the authors should show in C all bar graphs from all the conditions quantified from B, 100% included.

Reviewer #2

(Remarks to the Author)

The manuscript by Brianceau et al. titled “Continuous exchange of the inner ring component SctD is required for the assembly and function of the type III secretion system” investigates the dynamic exchange of SctD between the inner ring structure and a pool of membrane-diffusible units. While the authors present intriguing hypotheses regarding the biological significance of this exchange and its role in integrating the export apparatus component SctV, I have concerns that the experimental data do not support these conclusions.

Major concerns:

1. This manuscript builds on previous work from the same group (PMID: 33712575) in which they report that gastrointestinal pathogens, such as *Yersinia* and *Shigella*, prevent the premature unproductive secretion of type III secretion effectors at low pH through disassembly of the cytosolic complex formed by SctQ, SctK, and SctL; this disassembly only occurs at low pH, in other words, transition from neutral to low pH leads to disassembly of the cytosolic complex. In that study they showed that in *Yersinia*, low pH sensing occurs in the periplasm resulting in the surprising dissociation of the inner membrane structural component SctD. Importantly, the disassembly was reversible upon returning to neutral pH, allowing rapid activation of the system after passing through the acidic environment of the stomach. Indeed, this reversible mechanism triggered by the sensing of low pH was the central finding of the study.

In the current study, however, the authors propose that the exchange of SctD occurs under steady state, rather than pH-triggered, and that the exchange is required for the integration of the export apparatus component SctV. A thorough read of the methods shows that all the experiments were performed at neutral pH. Previously, under identical experimental conditions sctD at neutral pH was stably associated to the inner rings while in the current study it is not, and the observed shuttling occurs continuously at neutral pH. Therefore, these results are not compatible with their previous study. This discrepancy, however, was not addressed raising serious concern about the validity and/or rigor of the current studies.

2. A critical issue with this study is the inappropriate choice of methodology to address the central hypothesis, namely that the structural inner membrane protein SctD continuously shuttles in and out of the inner rings. To substantiate this claim, the authors rely on diffraction-limited imaging microscopy. Previous studies in which the senior author participated, have shown that “*Y. enterocolitica* injectisomes cluster within the bacterial membrane and that activation of the system leads to an increase in the number of injectisomes per cluster, rather than in the number of clusters per bacterium” (PMID: 25524451). The authors acknowledge this significant limitation in their methodology as a complicating factor in their analysis (lines 429-431), however this limitation does not just complicate the analysis but invalidates the study. There are multiple advanced techniques that enable optical imaging in super-resolution, and commercial microscopes are readily available for this purpose. Likewise, instead of conventional FRAP, the authors should have employed single-molecule FRAP (smFRAP), which combines the power of FRAP with super-resolution microscopy, allowing for recovery recordings below the diffraction limit (PMID: 27558844, 34007970). Without these techniques, it cannot be ascertained whether the newly formed fluorescent spots represent recovery at the original injectisome or at a newly formed injectisome present in the same cluster. Thus, with the experiments presented in the manuscript, it cannot be asserted that SctD is indeed shuttling in and out of the inner ring of a specific injectisome, thus invalidating the main conclusions of this study.

3. The authors attempt to establish a direct functional link between the exchange of SctD subunits and the injectisome’s ability to secrete client proteins. Using the cryo-EM structure of the *Salmonella*’s injectisome, where SctD is assembled in the inner ring, they identify two residues in the *Yersinia* SctD molecule that are in close contact within two adjacent SctD molecules in the assembled injectisome. These residues are mutated to cysteine to be able to prevent the putative exchange of SctD subunits either reversibly, by playing with the redox conditions of the bacteria periplasm, or irreversibly by using a chemical crosslinker, and then they assess the ability of *Yersinia* to secrete. Upon observing that the type III secretion system is unable to secrete when disulfide bonds are formed, the authors conclude that this must be due to the inability of SctD to exchange under these conditions. However, alternative explanations for this phenotype must be considered. While the injectisome is a robust structure, it is not a static; subtle conformational changes are relayed from the tip, through the needle to the base and the cytosolic complex enabling activation, substrate switching, and appropriate client protein selection (PMID: 27277624, 21143311, 16227202, 31260457). The crosslinking of SctD may impede a crucial conformational change necessary for successful secretion. Therefore, the experiments conducted are not sufficient to support the conclusions.

4. The authors attempt to explain the necessity of SctD to exchange, by focusing on the export apparatus component SctV. They question how SctV can be integrated into the injectisome if the membrane-spanning rings are assembled beforehand. Traditionally, the type III secretion field, classified all inner membrane-predicted proteins not part of the inner rings as the export apparatus. However, it is now clear that SctV does not fit within the same structural category as the other four export apparatus proteins (SctRSTU). While SctRSTU do not exhibit a typical membrane protein topology and instead form a helical

structure largely located within the periplasm (PMID: 29967543), enclosed by the SctDJ rings, SctV is positioned outside the SctDJ rings as observed by cryo-ET (PMID: 31744874). Therefore, there is no necessity for the inner rings to disassemble to accommodate SctV into the structure. In fact, it has been previously demonstrated that while SctRSTU cannot be incorporated into preformed injectisomes, SctV can be readily integrated into pre-assembled type III secretion machines (PMID: 20876096). The observation that the strain containing the mutated cysteines secretes only under reducing conditions does not demonstrate that SctV failed to integrate into the structure. Rather, it most likely reflects an entirely unrelated phenotype, as discussed previously. Given that purified injectisomes lack SctV (further demonstrating that SctV is peripherally associated with the injectisome rather than being located within the inner rings), it is essential to directly visualize the injectisome in situ by cryo-ET. This approach is necessary to determine whether SctV is incorporated into disulfide-bridge-containing injectisomes or absent altogether.

In summary, the studies do not provide sufficient evidence to support the stated conclusions.

Version 1:

Reviewer comments:

Reviewer #1

(Remarks to the Author)

Manuscript ID: NCOMMS-24-57188A

Title: Continuous exchange of the inner membrane ring component SctD is required for the assembly and function of the type III secretion system

The authors have conducted an excellent study on the membrane ring component SctD. They convincingly demonstrate its continuous exchange and mobility at the inner membrane, a process that provides novel insights into the mechanistic model of injectisome assembly and appears to be crucial, even essential, for protein secretion through the T3SS in *Yersinia*. The experimental approaches employed in the study (i.e. secretion manipulation, reversible cross-linking, and FRA) offer a stepwise in vivo dissection of the system, establishing a powerful methodological pipeline for studying the molecular mechanism that underlies bacterial secretion.

Following extensive revision, the manuscript now presents a clear and logical demonstration of their findings. The experimental procedures are thoroughly described, the data are well presented, the figures and tables are well formatted with comprehensive legends that greatly enhance clarity. All those changes made their results robust and conclusive. Additionally, the authors have fully addressed all my previous concerns and comments with well-reasoned and thorough responses.

Following these revisions, I strongly recommend this manuscript for publication

Reviewer #2

(Remarks to the Author)

I appreciate the effort by the authors of the manuscript by Brianceau et al., titled "Continuous exchange of the inner ring component SctD is required for the assembly and function of the type III secretion system", to address the issues previously raised by this reviewer. However, the explanations provided fall short of being convincing. Bold claims, such as the idea that a core structural component of the injectisome, SctD, is capable of shuttling in and out of the very structure it forms part of, require a high standard of evidence that the authors have not provided. Since the original visualization of the injectisome in 1998, successive structural studies have provided increasingly detailed atomic-level insight into this remarkable bacterial nanomachine. A review of standard injectisome purification protocols reveals that the structure is robust enough to endure harsh biochemical handling without losing integrity. If a steady-state exchange of SctD subunits were occurring, this would likely have been detected during structural studies as a distinct conformational class, yet no such evidence exists. Moreover, SctD and SctC share an extensive hydrophobic interface that is critical for injectisome assembly. The authors do not explain how SctD could continuously exchange without destabilizing this interface, and, by extension, the entire structure. The authors refer to "re-binding of SctD to the injectisome," but this is a mischaracterization: SctD does not bind to the injectisome, it is a core structural component of the injectisome. Without SctD, there is no injectisome. While disassembly of the cytosolic complex formed by SctQ, SctK, and SctL is conceptually easy to accept, since these are not core structural components and are routinely lost during purification, claiming that SctD exchanges within a functional injectisome demands far stronger evidence. In response to my previous critique, that all experiments in the current study were performed at neutral pH, despite the authors' earlier work showing that disassembly occurs at low pH to prevent unproductive secretion in the stomach, the authors argue that "at neutral pH 8% of SctD molecules were mobile." However, Figure 2b shows that upon induction of SctD in trans, almost all membrane-localized puncta formed by EGFP-SctD disappear, suggesting far more extensive exchange than 8%. This observation is not clearly interpreted by the authors, and alternative explanations are equally

plausible. Considering what is known about the structural rigidity of the injectisome, it is unclear why this hypothesis is being entertained in the first place.

The central issue remains the methodology used to support the key claim: that “the structural inner membrane protein SctD continuously shuttles in and out of the inner rings.” To substantiate this, the authors rely on diffraction-limited fluorescence microscopy, which has a spatial resolution limit of ~200 nm. Two objects must be separated by at least 200 nm to be resolved, yet within this space, 20–50 large proteins could fit side by side. Thus, the technique lacks the resolution to detect molecular-level exchange within such a compact macromolecular structure. While I understand the limitations of current imaging approaches, the claims made in the manuscript cannot be substantiated by the technique employed.

As I stated previously: extraordinary claims require extraordinary evidence, and diffraction-limited microscopy does not meet the standard required here.

Regarding the crosslinking experiments, where the authors attempt to establish a functional link between SctD exchange and secretion, they invoke “the principle of parsimony” or Occam’s Razor to argue that “the lack of secretion under crosslinking conditions, that is, when exchange cannot occur, is most likely due to the lack of exchange of SctD.” However, no direct demonstration of exchange is provided. We do know, however, that while the injectisome is robust, it is not static: subtle conformational changes propagate from the tip, through the needle, to the base and cytosolic complex, enabling activation, substrate switching, and client protein selection (PMIDs: 27277624, 21143311, 16227202, 31260457). By the same principle of parsimony, one could equally (and more plausibly) argue that the crosslinking of SctD may impede a crucial conformational change necessary for successful secretion.

The experiments shown in Figure 9 are presented as proof that “the exchange of SctD is important for the integration of the export apparatus into the assembled inner membrane ring structures.” However, these are secretion assays; they do not directly assess whether the export apparatus has been incorporated into the injectisome. Furthermore, the level of complementation shown in the blots (approximately 20% of wild-type levels) is low and suggests experimental issues. It is well established that the base of the injectisome forms via membrane insertion of the SctRSTU complex, and that SctRSTU cannot be incorporated into preformed injectisomes (PMID: 20876096). Indeed, it was shown that inner rings formed in the absence of the export apparatus exhibit a 23-fold symmetry, whereas fully assembled injectisomes consistently show 24-fold symmetry (PMID: 31744874), strongly indicating that export apparatus integration must precede inner ring assembly.

That same study also clearly demonstrated that membrane “lifting” occurs in the absence of SctV, and that this is a property of the SctRST complex. Therefore, it is not clear what the authors mean when they write: “Interaction with the SctV nonamer in the IM is the most likely candidate for lifting SctRST from the inner membrane.”

There also appears to be a misreading of the literature. The authors write: “This may explain why, as nicely shown in Wagner et al., PNAS 2010, doi 10.1073/pnas.1008053107, SctRSTU cannot integrate into the base in the absence of SctV.” On the contrary, Figure 3 of that paper clearly shows that SctR (SpaP in *Salmonella*) can be incorporated into the needle complex in the absence of SctV (InvA).

Due to the current difficulty in purifying SctV-containing injectisomes, the precise membrane positioning of SctV remains unresolved. Cryo-ET data suggest that SctV transmembrane domains may be located peripherally, possibly outside the SctD transmembrane ring (PMID: 31744874). What is clear is that SctV is only loosely associated with the injectisome and that, unlike SctRSTU, it can be functionally incorporated into pre-assembled complexes.

As such, the experiment added by the authors in Figure 9 does not convincingly support their model, nor does it substantively advance our understanding of export apparatus integration.

In conclusion, while the authors have made an effort to address the concerns raised, the core issues remain unresolved. The central claim, that SctD undergoes continuous exchange within the assembled injectisome, relies on insufficient evidence, is inconsistent with established structural data, and is not adequately reconciled with the authors’ own prior work. The methodologies employed are not capable of supporting the conclusions drawn, and several key interpretations of the literature appear either inaccurate or incomplete. Without more rigorous experimental support and clearer mechanistic insight, the study does not meet the standards of clarity, rigor, and reproducibility required for publication in *Nature Communications*.

Response to reviewer comments

Note to reviewers: For your information, we have summarized the main changes to the figures and supplementary figures below. This overview also shows the renumbering of supplementary figures resulting from the inclusion of new material.

Figure number	Previous number	Description
1	1	Minor changes (clearer labels)
2	2	Integration of the previous Suppl. Fig. 2, visual description of experimental workflow
3	3	(no changes)
4	4	More quantitative immunoblot method, improved labeling of blots
5	5	(no changes)
6	6	Complete quantification of secretion efficiency
7	7	Complete quantification of secretion efficiency
8	8	(no changes)
9	9	Addition of new results for core export apparatus component SctS
S1	new	Visual overview of the position of native and inserted cysteines and mutations in the SctD(L4) and SctD(2C) variants in the SctD sequence and structure
S2	S1	(no changes)
S3	new	Overview and explanation of experimental setup and conditions for the experiments in the main figures
S4	S3	(no changes)
S5	new	Cellular protein levels of the EGFP fusion proteins over the time range of the experiments
S6	new	Additional panels highlighting the positions of relevant amino acids in the structural model of the YeSctCDJ complex
S7	new	New experiments showing the absence of crosslinking of SctD(2C) with its periplasmic interacting partners SctC and SctJ
S8	S4	(no changes)
S9	S5	(no changes)
S10	S6	(no changes)
S11	S7	(no changes)
S12	S8	(no changes)

Reviewer #1 (Remarks to the Author):

General comment: The authors have performed a wonderful study regarding the membrane ring component SctD and have proven its continuous exchange and mobility at the inner membrane, a process that seems to be important or even essential for protein secretion through T3SS in *Yersinia*.

The study provides a novel insight into the mechanistic model of injectisome assembly and the experiments performed are a major breakthrough in the field. The secretion process manipulation, the reversible cross-linking and the FRAP experiments presented in the paper provide a step-to-step dissection of the system *in vivo* that appears to be a powerful tool/pipeline to study bacterial secretion. The manuscript is well written with a nice flow, experimental procedures seem to be clear and the results robust and conclusive. However, some points the authors made need to be clarified better and/ or described in more detail, so that the reader is not confused or even mis-led. Additionally, some control experiments would be essential in some cases so to strengthen their point.

Overall, according to my opinion, this manuscript should be accepted for publication, if the authors address the major and minor comments listed below.

We thank the reviewer for the insightful and positive feedback on the manuscript. We have implemented the suggestions for improvement as detailed below.

Major comments:

1: My first major comment is about Figure 2 and Figure S2. Unfortunately, I found it very confusing and although I understand the purpose and its importance, I fail to follow it either in the text and or in the figure. According to my opinion, the authors should re-design Figure 2 and perhaps merge it with S2 as the data presented in S2 are essential to support the manuscript. Of course, the manuscript will need revision to match the new Figure 2.

As an example, in Figure 2A, the authors show one micrograph on the right and mention "wt". What is this wt stands for? No plasmid in those cells? It contains the plasmid but there has been no addition of arabinose? Has it grown in parallel, collected and analysed at the end of the experiment? A clear description is required. In Figure S2A similar question.

Additionally, in Figure S2, what do the micrographs represent? The (-) and (+) images both shown foci. Do they come from the same cultures as from Fig. 2A with no arabinose added and incubated for the same time? Or is the cultures before adding arabinose? If the latter case is correct, I found those images redundant since you have the "wt" micrograph and they are shown in Fig. 2. I suggest the authors to redesign the figures and write more descriptive legends.

We apologize for the lack of clarity in this set of figures, and have changed them based on the feedback from the reviewer.

As suggested, we have merged Fig. 2 and S2. The new Fig. 2 now includes all relevant images and as a result shows the difference between the micrographs for EGFP-SctD and EGFP-SctQ. While EGFP-SctD is replaced by SctD(WT) and SctD(L4), both of which are integrated into the assembled injectisomes, EGFP-SctQ still binds to the injectisome in presence of additional SctD(WT), but not SctD(L4), as the latter cannot bind SctQ. This provides strong additional evidence for the exchange of SctD in fully assembled injectisomes.

To explain the experiment, we included a sketch of the timeline of the experiment, showing at which timepoints the micrographs were taken, and clearly labeled the micrographs taken before and after induction of expression of additional SctD(WT/L4) from plasmid. We also more clearly label the micrographs with and without induction of expression from plasmid (" + arab." / " no arab."). Finally, as suggested by the reviewer, we only show one image of the controls imaged before the induction of secretion.

We also completely rewrote and extended the figure legend, and the description of the results in the main text (lines 152-163, all line numbers refer to the manuscript version with the highlighted changes). While the figure is still complex, due to the inclusion of the prior Fig. S2, we think that the results are presented in a much clearer and more accessible way now.

2: There is some discrepancy regarding SctQ membrane localization and efficient exchange. When the authors used SctD (L2) mutant that has low/ no exchange, SctQ exchange is also affected. They explain this phenotype due to the lack of SctK interaction with SctD. Later in the text, when using SctD (2C) mutant that again shows no SctD membrane exchange, but SctQ one is not affected. This is because SctD is immobilized in the membrane and we make the assumption that SctK is not affected from that? The authors should elaborate all options, and discuss a little bit further those two different results. How those two different results are justified? Does SctQ need the SctD to be mobile or not?

Thanks for mentioning this potential source of misunderstanding. In general, presence of SctD is required for binding of SctQ to the injectisome, as was shown in several studies, perhaps most directly in Diepold et al., EMBO J, 2010 (doi 10.1038/emboj.2010.84).

The two variants of SctD used in the manuscript differ in their effect on SctQ:

The SctD(L4) variant used in Fig. 1 and 2 has a mutation in the cytosolic FHA domain and a dominant negative effect on T3SS secretion when co-expressed with native SctD (Gamez et al., J Bact, 2012, doi 10.1128/JB.00513-12). Mutations in the FHA domain of SctD can lead to loss of binding of SctK and/or SctQ to SctD (Tachiyama et al., JBC, 2019, doi 10.1074/jbc.RA119.009125). We show in Fig. 1 and 2 that the expression of SctD(L4) after assembly of the injectisome prevents secretion and binding of SctQ to the injectisomes, strongly indicating that it is integrated into existing pre-assembled injectisomes. Notably, we do not use this mutant for exchange assays.

In contrast, the SctD(2C) variant was designed by us to be able to crosslink two neighboring SctD monomers in the ring structure. This mutant variant does not affect the binding of SctQ to the injectisome under any condition tested, as shown in the micrographs of the Suppl. Fig. 11 (previously Suppl. Fig. 7). Moreover, it is fully active for secretion in its non-crosslinked state, in line with SctQ binding, which is required for secretion. All exchange and crosslinking assays in Fig. 4-8 were performed with this strain. This includes the experiment for Fig. 8, where we tested whether exchange of SctD is required for exchange of SctQ, which is not the case.

In summary, the presence of SctD is required for SctQ binding to the injectisome in general. SctD(L4) prevents SctQ binding to the injectisome, while SctD(2C) does not affect it.

The exchange of SctD shown in this manuscript is not required for SctQ exchange, as shown in Fig. 8.

While we already use a color code to distinguish the different SctD versions in the manuscript, we now describe both of the variants, the effect on secretion, and the rationale for using them in the respective experiments more clearly at the respective positions in the manuscript (lines 90-109, 152-163 for L4 and lines 256-258, 272-274 for 2C). We also included a **new Suppl. Fig. 1** highlighting the respective mutations in both variants and mention it in the main text.

3: Another section needs further clarification, the cross-linking experiments presented in Figure 4. As a start, it seems that the wt cysteines are present in the SctD (2C) protein. Is that correct? If so, it should be clearly mentioned in the text and explain why the native cysteines were retained. Additionally, one would expect that those cysteines should also interact and cross-link with something in close proximity, especially when non-specific BM(PEG)2 was used. How do the authors justify this result? Why those native cysteine are not reactive?

While SctD contains additional cysteines, these are all in the cytosolic domain of the protein and none is naturally present in the periplasmic domain. We now show this in a schematic way in the **new Suppl. Fig. 1**. Given that the reduction of the medium by addition of DTT affects the periplasm, but not the cytosol, and that the crosslinker BM(PEG)₂ is not membrane-permeable, the cytosolic cysteines will not be affected (in line with the finding that secretion is not reduced for wild-type-SctD, Fig. 6A).

Looking at this question in a wider context and also considering cysteine residues in SctC and SctJ (the interacting partners of SctD in the periplasm), the only cysteine that might be affected is Cys-19 of SctJ; however, it is still too far from the neighboring Cys-19 and from the two periplasmic cysteines in SctD to be crosslinked by BM(PEG)₂, let alone by native oxidative crosslinking. To make these points clearer, we now included a **new Suppl. Fig. 6** on the structure and modelling of the respective residues and refer to it in the main text (line 273-274).

More directly, we also excluded a crosslinking of SctD to SctC or SctJ by an additional experiment, see response to next point.

#4: Regarding the cross-linking experiments, one would also expect many more and of high-molecular weight cross-linking products, having in mind that SctD is forming 24mer ring-like structures at the membrane, interacts with SctK, SctJ, SctV and SctC rings. How the authors secure that the in vivo cross-linking results shown in Figure 4E are only due to SctD-SctD intra-molecular cross-linking? Why SctD could not interact with other components that they are in close proximity/interacting at the membrane? It is possible that those products are not easily analysed on gel, i.e. due to size (too big to resolve in the gel). Do the authors observe protein precipitation in the well of the gels? If yes, this should be discussed. MD simulations combined with analysis of structural models indeed can predict interaction sites and no one argues against the validity of those experiments. However, taking the current results and according to the protein size shown in the gel, someone can argue that SctJ-SctD and/or SctD-SctV complexes could be the cross-linked products detected. To strengthen their results, the authors should use an antibody against other injectisome proteins (i.e. SctV, SctJ) in Figure 5E, so to verify no cross-linking between those components and SctD.

This is an important statement to be made firmly and clarified, since the rest of the paper relies on the fact that SctD protomers cannot be exchanged and this intra-protomeric dynamic exchange is essential for T3SS to function properly.

This is indeed an important point, which we addressed in the revision. We repeated the experiments (using a different membrane treatment to exclude that smaller membranes are transferred through the membrane during the extended transfer times used to ensure transfer even of higher-molecular weight complexes). Importantly, in these experiments, we used not only a FLAG antibody to detect Flag-SctD as previously, but also antibodies against SctC and SctJ, and included controls to show the binding specificity of these antibodies. As for complexes/precipitates not entering the gel, we see a clear band for SctC, which likely corresponds to previously observed SDS-resistant secretin rings (Kowal et al., Structure, 2013, doi 10.1016/j.str.2013.09.012), but not for SctD or SctJ. More importantly, the results, displayed in the **new Suppl. Fig. 7**, show that neither SctC, nor SctJ are part of the ~140 kDa and >180 kDa bands seen in the SctD immunoblot upon crosslinking.

5: Finally, my main concern is about the final statement the authors make regarding the integration of the export apparatus and how this depends on SctD exchange. I think that the experiments performed are not sufficient to make such a firm statement. Perhaps it can be suggested or hypothesised but more experiments are needed in order to prove their point.

In more detail, the in vivo secretion results in Figure 9 show hardly any complementation, if one is to compare the wt profile with the complemented strain (comparing lane 1 with lanes 4-5), suggesting that

the system is compromised by those gene deletions and cannot be properly restored. I think the authors should make a comment on the text.

Moreover, if we focus and compare lanes 7 and 8, indeed there is a more prominent defect in secretion under oxidised conditions and this *in vivo* secretion defect is directly correlated by the authors with the lack of SctV integration at the export apparatus in the membrane. This is a statement that cannot be extrapolated firmly from *in vivo* secretion results. The authors present no physical evidence regarding SctV membrane integration, or even membrane localization and how it might be effected by SctD. SctV is the major component of the export apparatus and plays a major role in injectisome assembly, ATPase complex stabilization and secretion hierarchy. One could argue that the rest of the injectisome is affected or even downstream events are compromised in the absence of SctV *per se*.

I think the authors should perform a similar experiment using Δ sctJ or Δ sctC or Δ sctU cells as a control to show that under those conditions the secretion is restored and cross-linked SctD has no defective role in secretion through the system. Additionally, a FRAP experiment like the one performed for SctQ (Figure 8) should be performed for SctV to show no recovery after photo-bleaching is detected when SctD (2C) is cross-linked. Those results would make their statement more robust.

Otherwise, the authors with the current state of the manuscript should lower the strength of their conclusions all across the text. i.e lane 381: "... SctD allows for the integration..." could be re-phrased like could allow, we speculate it allows. Same for lane 457: "...our data unambiguously..." I think that unambiguously is too strong for that statement and they should consider re-phrasing and/or deleting, and authors could explore other possible conclusions in the discussion.

It is absolutely correct that compared to the wt control, where SctV (or, in the revised version, SctS, see below) are expressed at the same time as the remaining T3SS components, the secretion is strongly reduced, to about 20%, when SctV/SctS are expressed afterwards. However, taking this into account, we see a striking almost black-and-white picture when comparing the complementation in the SctD(WT) and SctD(2C) strain: While both strains show almost identical secretion levels under reducing conditions (lanes 5 and 8 of the gel), secretion in the SctD(2C) strain is completely abolished under native oxidizing conditions (lane 7), whereas secretion in the SctD(WT) strain is unaffected (lane 4), arguing against a more general effect of the temporary absence of SctV as such.

The idea for the FRAP experiment for SctV is excellent; however, complementation of T3SS component deletions from plasmid always requires higher expression levels of the respective proteins (this is true for all tested components, even when expressed at the same time). For complementations with fluorescently labeled components, this leads to an increased background level (most pronounced for membrane proteins such as SctV), which prevents FRAP experiments.

However, to test this phenotype for another export apparatus component, we have performed the equivalent experiments in Δ sctS strains, which we created for the revision, complemented with SctS expressed *in trans*. We also changed the quantification of the band intensity on the Coomassie-stained gels from horizontal scans, to more precise vertical line scans for each lane, which are more robust towards different background intensities. The results for SctS and SctV are highly comparable, strengthening the interpretation of a role of SctD dynamics in export apparatus integration.

We now show and discuss the new results, but at the same time more clearly mention the clearly reduced secretion in comparison to the wt control. As suggested by the reviewer, we now present the role of SctD dynamics in integration of the export apparatus into the SctDJ membrane rings as a hypothesis (albeit an attractive one), in line with the existing SctV complementation and the new SctS complementation results. Specifically, we changed "...these data show..." to "...these data are in line with the hypothesis..." (line 452, previously 381), and "...unambiguously shows..." to "...supports the hypothesis" (lines 558-559, previously 457).

Minor comments:

1, line 29: ...expressed at 37°C... A “C” is missing

We corrected this mistake and made sure to consistently use “°C” throughout.

2, line 76: ...as previously thought. I think a reference is missing here

We now list two reviews (including one of our own) for the previous view of the IM rings as static structures as references: Notti and Stebbins, *Microbiol Spectr* 2016, doi 10.1128/microbiolspec.VMBF-0004-2015; Diepold and Wagner, *FEMS Microbiol Rev* 2014, doi 10.1111/1574-6976.12061.

3, line 163: ...with kinetics slightly slower... According to my opinion ‘slightly’ should be removed because it gives the impression of fast exchange. Since SctD exchange is 3 times slower compared to SctQ and 2 times faster compared to SctV, one could argue and say that SctD exchange is slightly faster than SctV.

Done as suggested.

4, line 192: classical molecular dynamics... I think a reference is missing here

We now list the reference the approach is based on, and specifically refer to the Methods, where it is described in more detail, at this point.

#5: line 207-217: Here the authors are describing the cross-linking experiment. I am wondering why there are so low signals of the monomeric SctD especially in the non-cross-linked lane of the SctD wt and 2C. Could it be that the protein is degraded or even aggregating and not resolved in the gel? Could that be an indication of high molecular species formed? I think the authors should elaborate more on this result and justify it better on the text.

We also wondered about this observation, especially regarding that the overall SctD intensity in the different lanes appeared different. To ensure that smaller proteins do not transit the membrane during the longer transfer time we used to allow for the efficient transfer of the larger multimers, we used a higher methanol concentration in the blotting buffer (as mention above for #5). Indeed, this strongly increased the intensity of the lower molecular weight bands, while still allowing for the transfer of the higher MW bands. Using these settings, the overall band intensity is also very similar between the different conditions, indicating a more uniform transfer than previously. We have now added a representative blot in Fig. 4e.

6, line 581: using Fiji... the word software is missing here. Perhaps also the link of Fiji.

We have added “software” throughout and included the link, in addition to the reference, at the first mention.

7, line 581: How data were plotted and normalized is not described. Percentage of the wt? What is considered wt? etc. I think this information should be also written in this section.

We now describe the used densitometry and normalization in this section (lines 687-692).

8: All across the text, the authors mention different bacterial growing conditions in which they manipulate the system and can have protein secretion or not, injectisome assembly or not, etc. For

nonYersinia experts, it will be difficult to immediately connect the experimental conditions with the results shown, so to follow the flow of the text.

Could the authors consider adding a table listing the different conditions of bacterial growth (incubation time, temperature, medium composition) in supplement or in materials and methods and correlate it with T3SS step they monitor? That would provide some extra clarity to the text.

We thank the reviewer for this helpful suggestion. We now include a **new Suppl. Fig. 3**, which shows and explains the timeline and settings for all main experiments and hope that this will make the experiments and findings more easily accessible. This Suppl. Fig. is referenced in the legend of Figure 1.

9: The authors should use the same font and size all across the Figures. For example: in Figure 6 the numbers in A are smaller than in B; in Figure 7 the same and in Figure 9 although the numbers are the same size all across the figure, they are smaller than the rest of the figures.

We have adapted the figures as suggested and thank the reviewer for the notification.

10: It would have been easier for the reader if western blots are directly associated with the protein tracked and the antibody used. In that case, one won't have to go through the text or Figure legend to find this information. i.e. In Fig. 4E add anti-Flag- SctD and in 7B add anti-Flag-YopE.

We have added this information for all figures, including all supplementary figures.

11: In Figure 6B, the authors are comparing the secretion efficiency of Δ sctD complemented with SctD (2C) under oxidative or reduced conditions. In panel B though they show only two bar graphs. Additionally, in the legend is mentioned that secretion from wt was considered in each condition as 100%. Why is that? Why not to keep "wt oxidized" as 100% (physiologically relevant condition) and compare every other condition with this?

I think that the authors should show in B all the bar graphs from all the conditions quantified from A. Moreover, they should use one condition as 100% and express all the other ones as a percentage of this.

12: Same comment as above for Figure 7B and C. I think the authors should show in C all bar graphs from all the conditions quantified from B, 100% included.

We followed the advice and now show the quantification for all strains and conditions in Fig. 6 and Fig. 7 with "WT oxidized" as a reference.

Reviewer #2 (Remarks to the Author):

The manuscript by Brianceau et al. titled “Continuous exchange of the inner ring component SctD is required for the assembly and function of the type III secretion system” investigates the dynamic exchange of SctD between the inner ring structure and a pool of membrane-diffusible units. While the authors present intriguing hypotheses regarding the biological significance of this exchange and its role in integrating the export apparatus component SctV, I have concerns that the experimental data do not support these conclusions.

We thank the reviewer for the feedback and the constructive comments, which we have addressed in detail below.

Major concerns:

1. This manuscript builds on previous work from the same group (PMID: 33712575) in which they report that gastrointestinal pathogens, such as *Yersinia* and *Shigella*, prevent the premature unproductive secretion of type III secretion effectors at low pH through disassembly of the cytosolic complex formed by SctQ, SctK, and SctL; this disassembly only occurs at low pH, in other words, transition from neutral to low pH leads to disassembly of the cytosolic complex. In that study they showed that in *Yersinia*, low pH sensing occurs in the periplasm resulting in the surprising dissociation of the inner membrane structural component SctD. Importantly, the disassembly was reversible upon returning to neutral pH, allowing rapid activation of the system after passing through the acidic environment of the stomach. Indeed, this reversible mechanism triggered by the sensing of low pH was the central finding of the study. In the current study, however, the authors propose that the exchange of SctD occurs under steady state, rather than pH-triggered, and that the exchange is required for the integration of the export apparatus component SctV. A thorough read of the methods shows that all the experiments were performed at neutral pH. Previously, under identical experimental conditions sctD at neutral pH was stably associated to the inner rings while in the current study it is not, and the observed shuttling occurs continuously at neutral pH. Therefore, these results are not compatible with their previous study. This discrepancy, however, was not addressed raising serious concern about the validity and/or rigor of the current studies.

In the previous study mentioned by the reviewer (Wimmi et al., *Nat Comms*, 2021), we investigated the behavior of SctD (and the cytosolic components of the T3SS) at low external pH. Under these conditions, increased dissociation and/or decreased re-binding of SctD lead to an increased proportion of mobile SctD (48%) and a lower EGFP-SctD intensity at the injectisomes. This in turn causes the dissociation of the cytosolic T3SS components and suppression of secretion under these conditions. While at neutral pH, a much higher fraction of SctD was bound to the injectisomes, we found that 8% of SctD were mobile even under these conditions. The current project was inspired by this intriguing finding. As correctly stated by the reviewer (and mentioned in the Introduction, lines 62-65, and Materials and Methods, lines 626, 695, and 718 – all line numbers refer to the manuscript version with the highlighted changes), the current study is performed at neutral pH and looks at the exchange of SctD molecules in this physiologically relevant steady state.

Notably, this steady-state exchange does not lead to an overall reduction in average SctD binding – much as is the case for other exchanging proteins such as SctQ (Diepold et al., *PLOS Biol*, 2015, doi 10.1371/journal.pbio.1002039) or components of the nuclear pore complex (Rabut et al., *Nature Cell Biol* 2004, doi 10.1016/j.jmb.2018.06.039; see Tusk et al., *JMB*, 2018, doi 10.1016/j.jmb.2018.06.039, for a review). We mention and discuss the fact that protein exchange does not equate to incomplete injectisomes in the discussion (lines 516-519). The ongoing exchange of subunits described in this study is therefore well compatible to the findings in the previous study.

We amended the respective parts of the introduction to more clearly describe the above considerations (lines 59-65), and thank the reviewer for pointing out this possible source of confusion.

2. A critical issue with this study is the inappropriate choice of methodology to address the central hypothesis, namely that the structural inner membrane protein SctD continuously shuttles in and out of the inner rings. To substantiate this claim, the authors rely on diffraction-limited imaging microscopy. Previous studies in which the senior author participated, have shown that “*Y. enterocolitica* injectisomes cluster within the bacterial membrane and that activation of the system leads to an increase in the number of injectisomes per cluster, rather than in the number of clusters per bacterium” (PMID: 25524451). The authors acknowledge this significant limitation in their methodology as a complicating factor in their analysis (lines 429-431), however this limitation does not just complicate the analysis but invalidates the study. There are multiple advanced techniques that enable optical imaging in super-resolution, and commercial microscopes are readily available for this purpose. Likewise, instead of conventional FRAP, the authors should have employed single-molecule FRAP (smFRAP), which combines the power of FRAP with super-resolution microscopy, allowing for recovery recordings below the diffraction limit (PMID: 27558844, 34007970). Without these techniques, it cannot be ascertained whether the newly formed fluorescent spots represent recovery at the original injectisome or at a newly formed injectisome present in the same cluster. Thus, with the experiments presented in the manuscript, it cannot be asserted that SctD is indeed shuttling in and out of the inner ring of a specific injectisome, thus invalidating the main conclusions of this study.

As the reviewer states, injectisomes in *Y. enterocolitica* can form small clusters in the membrane. This is more pronounced in secreting bacteria, where the expression level of T3SS components is upregulated by about two-fold (Kudryashev et al., Mol Microbiol, 2014, doi 10.1111/mmi.12908). Notably, the increased expression and formation of clusters is a slow process, occurring over several hours and slowing down after about 120 min after induction of T3SS expression at 37°C (Kudryashev et al., 2014). Importantly, this increase and clustering occurs for all T3SS components, including SctV, for which there is no or very little recovery after photobleaching, as shown in Fig. 3, and is therefore extremely unlikely to be the cause for the observed recovery of SctD and SctQ.

We confirmed the above-mentioned findings for the expression levels of the T3SS components under the exact settings used for the experiments in this study for SctD and SctQ, whose levels increase by less than 10% per hour in the time range of the measurements of this manuscript (**new Suppl. Fig. 5**). This new production would lead to half-times of recovery of many hours, which is orders of magnitude slower than the exchange shown in our experiments.

We thank the reviewer for the suggestion of smFRAP. However, this technique that has, to the best of our knowledge, less than ten PubMed entries, all for eukaryotic cells, and we do not see how this niche approach requiring highly specialized equipment would create relevant additional information for our manuscript, especially given that our FRAP data already looks at distinctly localized complexes (which is often not the case for the proteins and complexes in eukaryotic membranes where smFRAP was used).

Finally, and in line with all of the above, we would like to point out that we show the exchange of SctD not only by FRAP, but also by functional assays independent of any fluorescence microscopy or labeling in Fig. 1 and 2.

We now refer to the new Suppl. Fig. 5 and the main points outlined above in the respective sections of the results part (lines 208-211) and the discussion (lines 489-496).

3. The authors attempt to establish a direct functional link between the exchange of SctD subunits and the injectisome’s ability to secrete client proteins. Using the cryo-EM structure of the Salmonella’s injectisome, where SctD is assembled in the inner ring, they identify two residues in the Yersinia SctD molecule that are in close contact within two adjacent SctD molecules in the assembled injectisome. These residues are mutated to cysteine to be able to prevent the putative exchange of SctD subunits

either reversibly, by playing with the redox conditions of the bacteria periplasm, or irreversibly by using a chemical crosslinker, and then they assess the ability of *Yersinia* to secrete. Upon observing that the type III secretion system is unable to secrete when disulfide bonds are formed, the authors conclude that this must be due to the inability of SctD to exchange under these conditions. However, alternative explanations for this phenotype must be considered. While the injectisome is a robust structure, it is not a static; subtle conformational changes are relayed from the tip, through the needle to the base and the cytosolic complex enabling activation, substrate switching, and appropriate client protein selection (PMID: 27277624, 21143311, 16227202, 31260457). The crosslinking of SctD may impede a crucial conformational change necessary for successful secretion. Therefore, the experiments conducted are not sufficient to support the conclusions.

In the manuscript, we directly show the influence of crosslinking on exchange of SctD and on T3SS effector secretion. Given the localization of the two chosen amino acids and the very specific action of crosslinking on these amino acids, the principle of parsimony indicates the lack of exchange as the most likely (and straightforward) explanation for the lack of secretion under these conditions. We agree with the reviewer that as for most experiments, other hypothetical, more indirect explanations are possible. While a signaling pathway such as the one shown by the references listed by the reviewer for *Shigella* and *Salmonella* (i) has not been shown for *Yersinia*, (ii) this signaling pathway (needle tip – needle – export apparatus) does not involve SctD, and (iii) the likelihood that crosslinking – both the reversible native crosslinking and the crosslinking with relatively long and flexible crosslinker BM(PEG)₂ – of the two selectively introduced Cysteine residues affects this pathway in particular is very low, we now mention alternative explanations and the above considerations in the discussion (lines 551-558).

4. The authors attempt to explain the necessity of SctD to exchange, by focusing on the export apparatus component SctV. They question how SctV can be integrated into the injectisome if the membrane-spanning rings are assembled beforehand. Traditionally, the type III secretion field, classified all inner membrane-predicted proteins not part of the inner rings as the export apparatus. However, it is now clear that SctV does not fit within the same structural category as the other four export apparatus proteins (SctRSTU). While SctRST do not exhibit a typical membrane protein topology and instead form a helical structure largely located within the periplasm (PMID: 29967543), enclosed by the SctDJ rings, SctV is positioned outside the SctDJ rings as observed by cryo-ET (PMID: 31744874). Therefore, there is no necessity for the inner rings to disassemble to accommodate SctV into the structure. In fact, it has been previously demonstrated that while SctRSTU cannot be incorporated into preformed injectisomes, SctV can be readily integrated into pre-assembled type III secretion machines (PMID: 20876096). The observation that the strain containing the mutated cysteines secretes only under reducing conditions does not demonstrate that SctV failed to integrate into the structure. Rather, it most likely reflects an entirely unrelated phenotype, as discussed previously. Given that purified injectisomes lack SctV (further demonstrating that SctV is peripherally associated with the injectisome rather than being located within the inner rings), it is essential to directly visualize the injectisome in situ by cryo-ET. This approach is necessary to determine whether SctV is incorporated into disulfide-bridge-containing injectisomes or absent altogether.

In summary, the studies do not provide sufficient evidence to support the stated conclusions.

As the reviewer correctly points out, recent structural findings show that SctRST form a tight complex that is located above the continuation of the inner membrane and forms a pseudohelix at the base of the rod and needle, and that SctU interacts with SctRST. Interaction with the SctV nonamer in the IM is the most likely candidate for lifting SctRST from the inner membrane. This may explain why, as nicely shown in Wagner et al., PNAS 2010, doi 10.1073/pnas.1008053107, SctRSTU cannot integrate into the base in the absence of SctV. We therefore agree that SctRSTUV cannot be seen as a monolithic complex. On the other hand, it was found that the complete export apparatus can assemble independently of any other T3SS

components (in *Salmonella*, Wagner et al., PNAS 2010, doi 10.1073/pnas.1008053107, as well as *Yersinia*, Diepold et al., Mol Microbiol 2011, doi 10.1111/j.1365-2958.2011.07830.x). Perhaps more importantly, all current structural evidence suggests that SctV is inside the SctD ring (more precisely, the SctV TMHs are inside the ring of SctD TMHs), rather than outside, and to the best of our knowledge and based on many discussions, this view is shared across the field. We have included some images from recent publications of leading structural labs, including the reference listed by the reviewer (Butan et al., PNAS 2019, PMID 31744874, doi 10.1073/pnas.1916331116) below.

Figure for Review: Structural models of the export apparatus in the context of the assembled T3SS.

From left to right, from Soto et al., PNAS 2022; doi 10.1073/pnas.2218010119; Butan et al., PNAS 2019 (SctV in purple, SctD in turquoise); Kuhlen et al., PLOS one 2021, doi 10.1371/journal.pone.0252800 (SctV in red).

Nevertheless, to fully address the reviewer's comment and test an export apparatus component other than SctV, we have repeated the experiments with SctS. We complemented a Δ sctS strain in the SctD(WT) and SctD(2C) strain background, which we created for this revision experiment, with SctS from plasmid, in the same way as previously done for SctV. The results are remarkably similar and we included the additional data as part of **Figure 9 (new panels c and d)**.

In summary, we believe that as outlined above, the conclusions are supported by the data presented in the manuscript and that the additional experiments, considerations and clarifications added in response to the reviewer comments further strengthen the manuscript.

Reviewer #1 (Remarks to the Author):

The authors have conducted an excellent study on the membrane ring component SctD. They convincingly demonstrate its continuous exchange and mobility at the inner membrane, a process that provides novel insights into the mechanistic model of injectisome assembly and appears to be crucial, even essential, for protein secretion through the T3SS in *Yersinia*.

The experimental approaches employed in the study (i.e. secretion manipulation, reversible cross-linking, and FRA) offer a stepwise *in vivo* dissection of the system, establishing a powerful methodological pipeline for studying the molecular mechanism that underlies bacterial secretion.

Following extensive revision, the manuscript now presents a clear and logical demonstration of their findings. The experimental procedures are thoroughly described, the data are well presented, the figures and tables are well formatted with comprehensive legends that greatly enhance clarity. All those changes made their results robust and conclusive. Additionally, the authors have fully addressed all my previous concerns and comments with well-reasoned and thorough responses.

Following these revisions, I strongly recommend this manuscript for publication.

We thank the reviewer for this very positive evaluation and the valuable recommendations in the earlier comments!

Reviewer #2 (Remarks to the Author):

I appreciate the effort by the authors of the manuscript by Brianceau et al., titled “Continuous exchange of the inner ring component SctD is required for the assembly and function of the type III secretion system”, to address the issues previously raised by this reviewer. However, the explanations provided fall short of being convincing.

Bold claims, such as the idea that a core structural component of the injectisome, SctD, is capable of shuttling in and out of the very structure it forms part of, require a high standard of evidence that the authors have not provided. Since the original visualization of the injectisome in 1998, successive

structural studies have provided increasingly detailed atomic-level insight into this remarkable bacterial nanomachine. A review of standard injectisome purification protocols reveals that the structure is robust enough to endure harsh biochemical handling without losing integrity. If a steady-state exchange of SctD subunits were occurring, this would likely have been detected during structural studies as a distinct conformational class, yet no such evidence exists. Moreover, SctD and SctC share an extensive hydrophobic interface that is critical for injectisome assembly. The authors do not explain how SctD could continuously exchange without destabilizing this interface, and, by extension, the entire structure. The authors refer to “re-binding of SctD to the injectisome,” but this is a mischaracterization: SctD does not bind to the injectisome, it is a core structural component of the injectisome. Without SctD, there is no injectisome. While disassembly of the cytosolic complex formed by SctQ, SctK, and SctL is conceptually easy to accept, since these are not core structural components and are routinely lost during purification, claiming that SctD exchanges within a functional injectisome demands far stronger evidence.

We appreciate the reviewer’s perspective. We fully agree that these points are important and have already commented on them in the revised version. However, the results of the experiments in our manuscript, which include a variety of different types of assays and many controls, consistently indicate exchange of SctD. In line with this, while SctC and SctD indeed share a hydrophobic interface (Hu et al., *Nat Microbiol* 2019, DOI 10.1038/s41564-019-0545-z), additional stabilizing interactions between neighboring SctD molecules are required for a functional connection, which can be overcome, for example, by a simple Alanine replacement of a Glu and Tyr residue in neighboring SctD molecules (Schraidt and Marlovits, *Science* 2011, doi 10.1126/science.1199358). We now mention this in lines 452-454 (all line numbers refer to the manuscript version with the highlighted changes). More generally, we now specifically include a summary of these structural arguments, as well as their reconciliation with our results, in lines 477-488 of the discussion. We hope that this addition to the overview of the experimental evidence in this and previous papers (now covering lines 464-493) will allow readers to draw their conclusions and, in the best case, design experiments to test our and previous data and interpretations in their own experiments.

In response to my previous critique, that all experiments in the current study were performed at neutral pH, despite the authors’ earlier work showing that disassembly occurs at low pH to prevent unproductive secretion in the stomach, the authors argue that “at neutral pH 8% of SctD molecules were mobile.” However, Figure 2b shows that upon induction of SctD in trans, almost all membrane-localized puncta formed by EGFP-SctD disappear, suggesting far more extensive exchange than 8%. This observation is not clearly interpreted by the authors, and alternative explanations are equally plausible. Considering what is known about the structural rigidity of the injectisome, it is unclear why this hypothesis is being entertained in the first place.

The loss of EGFP-SctD foci in Fig. 2b specifically occurs upon the overexpression of SctD, which is clearly shown in Suppl. Fig. 2. Given the much higher amount of overall SctD under these conditions (>10x times increased, based on the Western blot result shown in Suppl. Fig. 2), but an unchanged number of injectisomes, the percentage of unbound SctD increases, and the previously bound EGFP-SctD molecules exchange with the additional unlabeled SctD molecules. The resulting strong dilution of EGFP-SctD at the injectisomes is a consequence of SctD exchange, which is exactly the point of the experiment. To avoid any confusion, we now describe this rationale even more clearly in lines 97 and 124-134.

The central issue remains the methodology used to support the key claim: that “the structural inner membrane protein SctD continuously shuttles in and out of the inner rings.” To substantiate this, the authors rely on diffraction-limited fluorescence microscopy, which has a spatial resolution limit of ~200 nm. Two objects must be separated by at least 200 nm to be resolved, yet within this space, 20–50 large proteins could fit side by side. Thus, the technique lacks the resolution to detect molecular-level exchange within such a compact macromolecular structure. While I understand the limitations of current imaging approaches, the claims made in the manuscript cannot be substantiated by the technique employed. As I stated previously: extraordinary claims require extraordinary evidence, and diffraction-limited microscopy does not meet the standard required here.

Fluorescence recovery after photobleaching (FRAP), which we used to detect the exchange of SctD, does not require single-molecule resolution. It measures the recovery of fluorescence intensity within a defined, diffraction-limited region, independent of the exact number of fluorophores present in that area. By principle, FRAP can therefore measure exchange of proteins in this diffraction-limited region, irrespective of the number of molecules (e.g. 24 molecules of EGFP-SctD). The resolution of the microscopy experiments is therefore absolutely sufficient to show the exchange of SctD. We now specifically include this argumentation in lines 470-476 in the revised manuscript. Beyond this, as stated previously, our manuscript provides strong evidence for the exchange of SctD by various experiments using other methods, including microscopy-independent approaches.

Regarding the crosslinking experiments, where the authors attempt to establish a functional link between SctD exchange and secretion, they invoke “the principle of parsimony” or Occam’s Razor to argue that “the lack of secretion under crosslinking conditions, that is, when exchange cannot occur, is most likely due to the lack of exchange of SctD.” However, no direct demonstration of exchange is provided. We do know, however, that while the injectisome is robust, it is not static: subtle conformational changes propagate from the tip, through the needle, to the base and cytosolic complex, enabling activation, substrate switching, and client protein selection (PMIDs: 27277624, 21143311, 16227202, 31260457). By the same principle of parsimony, one could equally (and more plausibly) argue that the crosslinking of SctD may impede a crucial conformational change necessary for successful secretion.

See replies to points 1 and 2 above for evidence of exchange of SctD, which is the basis of our argumentation.

The experiments shown in Figure 9 are presented as proof that “the exchange of SctD is important for the integration of the export apparatus into the assembled inner membrane ring structures.” However, these are secretion assays; they do not directly assess whether the export apparatus has been incorporated into the injectisome. Furthermore, the level of complementation shown in the blots (approximately 20% of wild-type levels) is low and suggests experimental issues. It is well established that the base of the injectisome forms via membrane insertion of the SctRSTU complex, and that SctRSTU cannot be incorporated into preformed injectisomes (PMID: 20876096). Indeed, it was shown that inner rings formed in the absence of the export apparatus exhibit a 23-fold symmetry, whereas fully assembled injectisomes consistently show 24-fold symmetry (PMID: 31744874), strongly indicating that export apparatus integration must precede inner ring assembly. That same study also clearly demonstrated that membrane “lifting” occurs in the absence of SctV, and that this is a property of the SctRST complex. Therefore, it is not clear what the authors mean when they write: “Interaction with the SctV nonamer in the IM is the most likely candidate for lifting SctRST from the inner membrane.”

There also appears to be a misreading of the literature. The authors write: “This may explain why, as nicely shown in Wagner et al., PNAS 2010, doi 10.1073/pnas.1008053107, SctRSTU cannot integrate into the base in the absence of SctV.” On the contrary, Figure 3 of that paper clearly shows that SctR (SpaP in Salmonella) can be incorporated into the needle complex in the absence of SctV (InvA).

Due to the current difficulty in purifying SctV-containing injectisomes, the precise membrane positioning of SctV remains unresolved. Cryo-ET data suggest that SctV transmembrane domains may be located peripherally, possibly outside the SctD transmembrane ring (PMID: 31744874). What is clear is that SctV is only loosely associated with the injectisome and that, unlike SctRSTU, it can be functionally incorporated into pre-assembled complexes.

As such, the experiment added by the authors in Figure 9 does not convincingly support their model, nor does it substantively advance our understanding of export apparatus integration.

In response to the corresponding reviewer comment in the previous round, we included additional experiments showing that this phenotype can not only be detected for SctV, but also for the core export apparatus component SctS. As shown in Fig. 9, the complementation of a Δ sctS strain with SctS from plasmid yielded results that were remarkably similar to the corresponding results for SctV. As mentioned in the previous response and discussed in the revised manuscript, this indicates that, while different members of the export apparatus have different structural and functional roles, the apparatus as such is integrated into the SctDJ ring, in line with the structural evidence and the general view in the field. The reason for the lower level of secretion of both complemented strains is that these strains need to express the complementing proteins and integrate the substructure, while the WT continuously secretes throughout this time. We now explain this, with a reference to the experimental setup shown in Suppl. Fig. 3, in lines 423-424. With respect to PMID 20876096, we focused on Fig. 4 and 5 showing different intermediate levels of integration of SpaQRS and InvA into the export apparatus upon expression at different timepoints. As the reviewer correctly states, Fig. 3 shows efficient integration of SctR/SpaP into the base band in absence of SctV/InvA. We thank the reviewer for highlighting that PMID 31744874 shows (in Suppl. Fig. 7) that membrane lifting occurs in absence of SctU and SctV. We changed the statement in lines 455-457 of the manuscript accordingly to: “...the export apparatus, whose SctR₅S₄T part moves from the IM towards the periplasm and interacts with SctU and a SctV nonamer”.

In conclusion, while the authors have made an effort to address the concerns raised, the core issues remain unresolved. The central claim, that SctD undergoes continuous exchange within the assembled injectisome, relies on insufficient evidence, is inconsistent with established structural data, and is not adequately reconciled with the authors' own prior work. The methodologies employed are not capable of supporting the conclusions drawn, and several key interpretations of the literature appear either inaccurate or incomplete. Without more rigorous experimental support and clearer mechanistic insight, the study does not meet the standards of clarity, rigor, and reproducibility required for publication in Nature Communications.

We appreciate the reviewer's differing interpretation of the data presented in the manuscript, and discuss the underlying arguments in the manuscript. We respectfully refer to the responses above (and in the previous rebuttal), and, more importantly, the data of the experiments presented in our manuscript, which provide the experimental evidence for our interpretation, allowing readers to draw their own conclusions.